# GrandQC: A comprehensive solution to quality control problem in digital pathology

Zhilong Weng [1,18], Alexander Seper[2,18], Alexey Pryalukhin[3], Fabian Mairinger[4], Claudia Wickenhauser[5], Marcus Bauer[5], Lennert Glamann [5], Hendrik Bläker[6], Thomas Lingscheidt[6], Wolfgang Hulla[3], Danny Jonigk [7,8], Simon Schallenberg[9], Andrey Bychkov [10,11], Junya Fukuoka[10,11], Martin Braun[12], Birgid Schömig-Markiefka[1], Sebastian Klein[1], Andreas Thiel[13], Katarzyna Bozek [14,15,16], George J. Netto[17], Alexander Quaas[1], Reinhard Büttner [1] & Yuri Tolkach[1] ✉

Histological slides contain numerous artifacts that can significantly deteriorate the performance of image analysis algorithms. Here we develop the GrandQC tool for tissue and multi-class artifact segmentation. GrandQC allows for high-precision tissue segmentation (Dice score 0.957) and segmentation of tissue without artifacts (Dice score 0.919–0.938 dependent on magnification). Slides from 19 international pathology departments digitized with the most common scanning systems and from The Cancer Genome Atlas dataset were used to establish a QC benchmark, analyzing inter-institutional, intra-institutional, temporal, and inter-scanner slide quality variations. GrandQC improves the performance of downstream image analysis algorithms. We open-source the GrandQC tool, our large manually annotated test dataset, and all QC masks for the entire TCGA cohort to address the problem of QC in digital/computational pathology. GrandQC can be used as a tool to monitor sample preparation and scanning quality in pathology departments and help to track and eliminate major artifact sources.

Digital pathology is an ongoing transformation of pathology as a medical specialty. Importantly, digitization allows the application of image analysis algorithms that support pathologists and make their work significantly more effective. AI-based image analysis algorithms

for diagnostic tasks (e.g., tumor detection, grading, subtyping; classification of non-neoplastic diseases) and for advanced applications (prognosis and prediction of therapy response in oncology) can revolutionize, objectivize, and personalize medicine, especially in the field

[1]Institute of Pathology, University Hospital Cologne, 50937 Cologne, Germany. [2]Danube Private University, 3500 Krems an der Donau, Austria. [3]Institute of Pathology, University Hospital Wiener Neustadt / Danube Private University, 2700 Wiener Neustadt, Austria. [4]Institute of Pathology, University Hospital Essen, Essen, Germany. [5]Institute of Pathology, University Hospital Halle, Martin Luther University Halle-Wittenberg, Halle (Salle), Germany. [6]Institute of Pathology, University Hospital Leipzig, Leipzig, Germany. [7]Institute of Pathology, University Hospital Aachen, Aachen, Germany. [8]German Center for Lung Research (DZL), Biomedical Research in Endstage and Obstructive Lung Disease Hannover (BREATH), Hannover, Germany. [9]Institute of Pathology, University Hospital Charite, Berlin, Germany. [10]Department of Pathology Informatics, University Hospital Nagasaki, Nagasaki, Japan. [11]Kameda Medical Center, Tamogawa, Japan. [12]MVZ Pathology and Cytology Rhein-Sieg, Troisdorf, Germany. [13]MVZ Pathology Bethesda, Duisburg, Germany. [14]Institute for Biomedical Informatics, Faculty of Medicine and University Hospital Cologne, University of Cologne, Cologne, Germany. [15]Center for Molecular Medicine Cologne (CMMC), Faculty of Medicine and University Hospital Cologne, University of Cologne, Cologne, Germany. [16]Cologne Excellence Cluster on Cellular Stress Responses in Aging-Associated Diseases (CECAD), University of Cologne, Cologne, Germany. [17]Department of Pathology and Laboratory Medicine, Perelman School of Medicine at the University of Pennsylvania, Pennsylvania, USA. [18]These authors contributed equally: Zhilong Weng, Alexander Seper. ✉e-mail: yuri.tolkach@gmail.com

of oncology[1–5]. However, one significant bottleneck for AI algorithm implementation is the quality control (QC) problem (Fig. 1A). Multiple artifacts are present in virtually all histological slides digitized by modern scanning systems. These artifacts are related to tissue processing and sectioning, staining, and digitization itself (Fig. 1A). Every artifact type can lead to significant and often critical (from the clinical point of view) misclassifications, such as false positive or false negative tumor tissue detection. The AI algorithms often fail silently, as their architecture usually suggests classification into pre-defined classes without the possibility of "unknown" or "uncertain" classification.

Although several tools were developed earlier and even open-sourced for research use (e.g., HistoQC, PathProfiler, HistoROI), they all have significant limitations, such as using non-deep learning technology for development (offering only moderate accuracy results for artifact detection)[6], applicability only in one diagnostic domain[7,8], non-comprehensive nature (detection of only single artifact types)[9–12], and restricted access[13]. Several commercial tools were developed that can be used only in the context of proprietary digital pathology software (e.g. SlideQC/Indica Labs, Artifactdetect/ PathAI, Automated Quality Control/ Proscia, and some others) and are not accessible in a research context.

In this study, we develop a powerful tool for QC in digital pathology (GrandQC) that allows for high-accuracy, comprehensive preprocessing of whole-slide images (WSI) including tissue detection and multi-class artifact detection, for further application of downstream image analysis algorithms or digital case sign-out. We extensively validated the GrandQC and show how it can be effectively used as a QC benchmark for scanning systems and pathology institutes analyzing data from 19 different sites. We open-source GrandQC and a large manually annotated test dataset with artifacts. Additionally, we analyze histological slides from all cohorts of The Cancer Genome Atlas (TCGA) project concerning their quality and open-source all QC masks that can now be easily used in further projects by researchers worldwide.

## Results
### Algorithm development (GrandQC tool)
Using a high-quality, precisely manually annotated, large training dataset (Fig. 1B, C), two pixel-level semantic segmentation algorithms were developed (tissue detection and multi-class artifact detection; tissue vs. background annotations involved automation using QuPath instruments) for H&E-stained WSIs, comprising the two-module GrandQC tool (Fig. 2A). The GrandQC detects following artifacts: air bubbles, slide edge, out-of-focus regions, pen markings, tissue folds, foreign objects, and dark spots, as well as tissue without artifacts. For

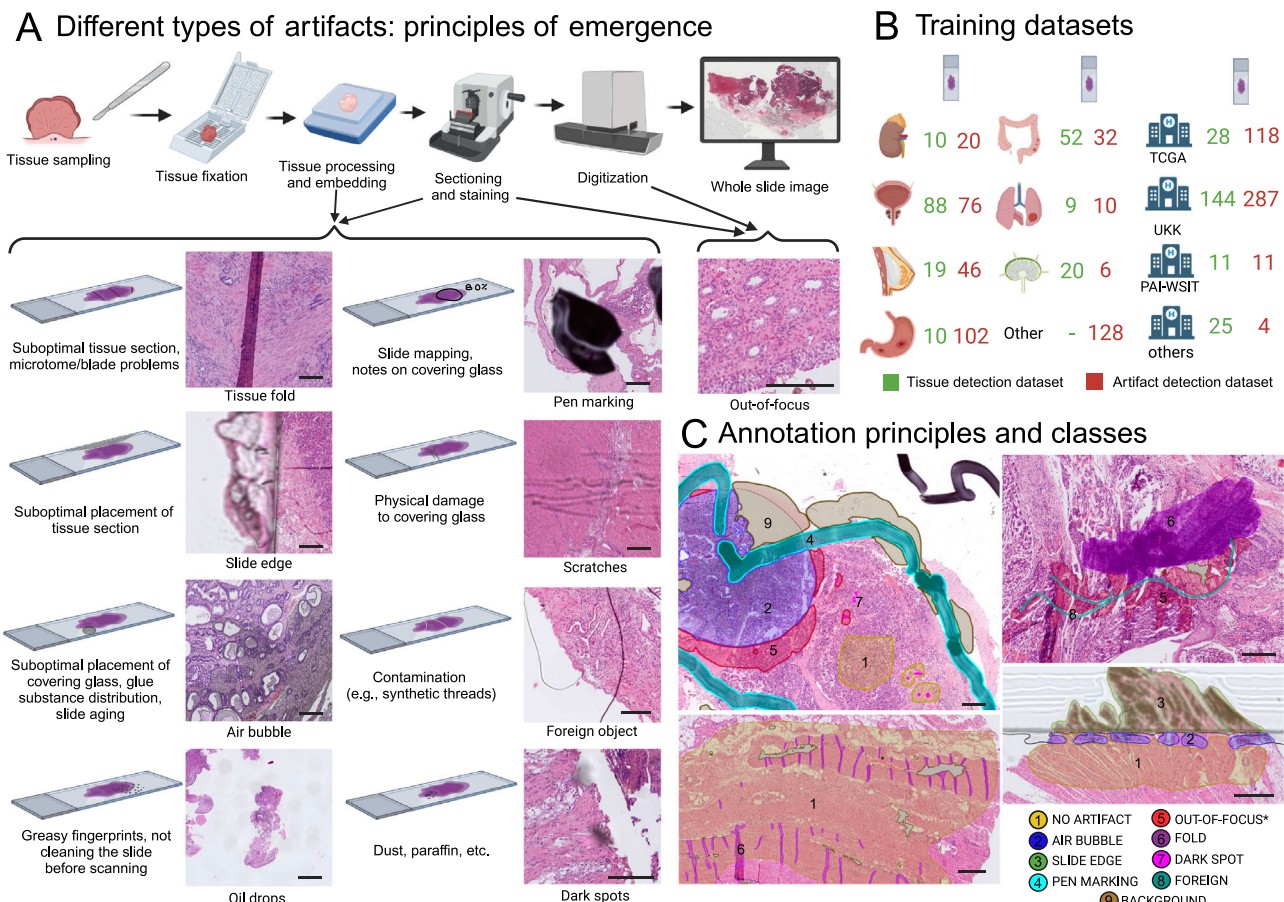

**Fig. 1 | Types of artifacts and training dataset preparation. A** Different types of artifacts: principles of emergence. Shown is a typical processing pipeline of pathology department from tissue sampling and submission by clinicians to histological slide preparation. The common mechanisms of artifact emergence are provided with most artifacts arising during preparation of the slides. The only digitization-specific artifact are out-of-focus regions which, however, might a consequences of suboptimal cutting and staining quality. **B** Training datasets. Two training datasets were prepared with partially overlapping cases: for tissue detection (slides $n = 208$) and artifact detection (slides $n = 420$) tasks. For large slide series included into the training the organs/tumor types are provided as well as source of slides. For details of datasets see Methods. **C** Annotations principles and classes. Precise manual annotations were performed by expert analysts concerning 9 classes shown (for training purposes AIR BUBBLE and SLIDE EDGE as well as DARK SPOT and FOREIGN bodies were merged as one class, correspondingly, due to similarity). Not shown are annotations for tissue detection tasks that included two classes (tissue and background). Abbreviations: TCGA – The Cancer Genome Atlas, UKK – University Hospital Cologne, PAI-WSIT – PAI-WSIT cohort. Scale bars in all microscopic images are 200 μm. Created in https://BioRender.com.

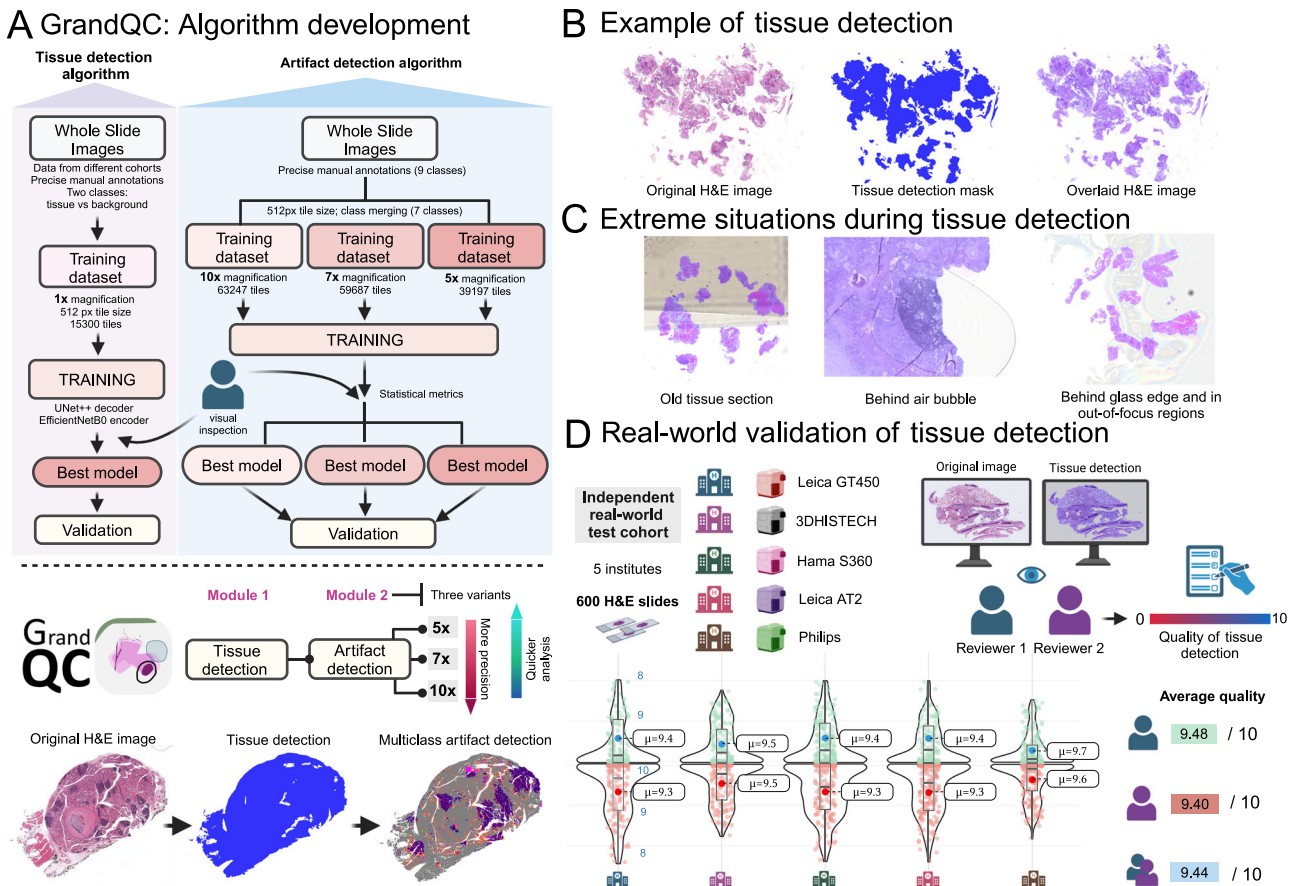

**Fig. 2 | GrandQC algorithm development and validation of tissue detection module. A** GrandQC: Algorithm development. Shown is the pipeline of the algorithm development for two separate modules: tissue detection and artifact detection. Both modules are pixel-wise segmentation networks with tissue detection working at 1x objective magnification level and artifact detection trained in three flavors for 10x, 7x, and 5x magnification with higher resolutions allowing more precision and lower resolutions quicker analysis at a cost of minimal changes in accuracy. Two modules build a tool: GrandQC. The tool is open-sourced for academic research use (https://github.com/cpath-ukk/grandqc). The principle of work is provided below. **B** Example of tissue detection in the biopsy case with multiple very small tissue particles showing reliable tissue segmentation. **C** Extreme situations during tissue detection. The algorithm performs very well in such situations as old tissue sections with poor quality of covering glass or glass edges, detection behind air bubble, glass edge our in out-of-focus regions. **D** Real-world validation of tissue detection. A heterogeneous real-world dataset containing 600 whole-slide images from different organs and types of specimens (5 pathology departments, 5 different scanning systems) was provided to two experienced human analysts that graded tissue detection on a scale 0–10 per slide. Single points were removed for overdetection with subtle, non-relevant underdetection grade as 7 points and any relevant tissue underdetection graded as low point number. Both analysts reported excellent tissue detection capabilities. With slide-level and average accuracy/quality (Reviewer 1: 9.48 of maximal 10 and Reviewer 2: 9.40 of maximal 10 points) results provided. All, mostly very fine inaccuracies were considered non-relevant. For the box plots in the figures, the center line represents the median, the red and the blue points represent the mean of the score, the box bounds depict the interquartile range (IQR), covering data from the 25th to 75th percentiles, which represents the middle 50% of the scores, the whiskers extend to a range of 1.5 times the IQR from the lower and upper quartiles, capturing a broader spread of the data. Created in https://BioRender.com. Source data is provided as a Source data file.

training purposes (Fig. 2A), we merged air bubble and slide edge artifacts as well as dark spot and foreign object artifacts into one class, respectively, due to their visual similarity. Moreover, the artifact detection algorithm is trained in three variants: for 5x, 7x, and 10x magnification (Fig. 2A). All three versions are accurate, but low-resolution versions allow quicker analysis in high-load situations at the cost of minimal accuracy loss. In the following, we extensively validate both tissue detection and artifact detection algorithms and suggest and evaluate a quality control benchmark for pathology institutes and scanning systems.

## Validation of tissue detection algorithm

Firstly, we performed a formal validation using segmentation accuracy metrics. For the test dataset of images (100 WSIs, enriched for different types of artifacts, different organs, Supplementary Methods), the average Dice score among the tissue/background classes was 0.957 (details to WSIs and performance in specimens from different organs in Supplementary Table 3). In a visual validation, the tissue detection

algorithm showed excellent performance and high pixel-level precision, especially in the biopsy setting (multiple very small particles; Fig. 2B) and in particularly difficult cases (Fig. 2C). Further, we performed a real-world validation that included 600 routine H&E-stained WSIs from five different departments (five different scanning systems) representative of different organs and specimen types (Fig. 2D). These were evaluated by two experienced analysts imitating real-world lab practice, who graded tissue detection quality using a scale of 0–10, while any relevant tissue underdetection was defined as a score of 6 or less (not evident in the validation). The average score for all cohorts and slides from the two graders was 9.44 out of 10 possible points (details in Fig. 2D), with human analysts reporting nearly ideal tissue detection quality.

## Validation of artifact detection algorithm

The WSI analysis pipeline of the GrandQC tool consists of two modules: tissue segmentation and artifact segmentation. An example of WSI processing with the artifact detection algorithm (which works in

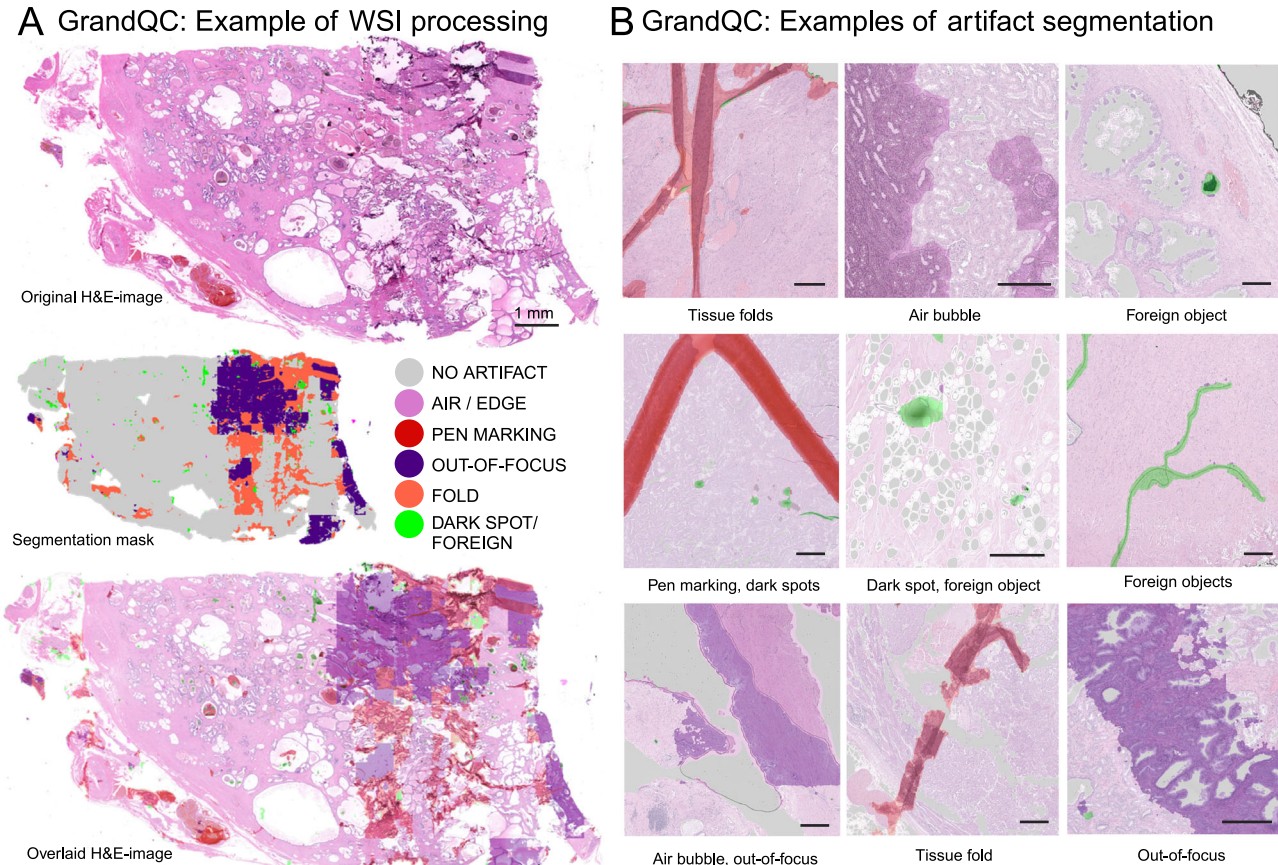

**A** GrandQC: Example of WSI processing

Original H&E-image

1 mm

Segmentation mask

- NO ARTIFACT
- AIR / EDGE
- PEN MARKING
- OUT-OF-FOCUS
- FOLD
- DARK SPOT/ FOREIGN

Overlaid H&E-image

1 mm

**B** GrandQC: Examples of artifact segmentation

Tissue folds | Air bubble | Foreign object

Pen marking, dark spots | Dark spot, foreign object | Foreign objects

Air bubble, out-of-focus | Tissue fold | Out-of-focus

**Fig. 3 | Examples of GrandQC application to whole-slide images. A** Example of processing of whole-slide image by GrandQC (7x version). For demonstration purposes an image from a prostatectomy case shown with substantial number of artificats. The detectable artifact classes as well as areas without artifacts are shown in different colors (see color legend). Background is shown as white color (detection by tissue detection module). **B** Further representative, high-resolution examples of artifact detection by GrandQC artifact detection module. All scale bars are 200 μm. Created in https://BioRender.com.

the segmented tissue regions) and its visualization is presented in Fig. 3A, with further examples of multi-class artifact detection in Fig. 3B (comparative for 5x, 7x, and 10x versions in Supplementary Fig. 3-6). For comprehensive statistical performance evaluation of the artifact detection algorithm, we created a large test dataset (details in Methods and Fig. 4A) with 318 slides manually and precisely annotated for different artifacts (including OOF), resulting in a large test dataset comparable to the training dataset in terms of the annotated area.

During the formal validation, all three artifact detection algorithm versions (5x, 7x, 10x) showed excellent segmentation accuracy of the tissue without artifacts (Dice score 0.919–0.938), with most Dice score fluctuations and lower for artifact classes related to inter-artifact misclassifications (here, the segmentation of tissue without artifacts allows for better estimation of overall performance of the tool; details in Fig. 4B).

**Runtime performance analysis**
Both modules of the GrandQC tool are relatively shallow encoder/decoder networks, requiring only ~1.5 GB of GPU RAM and having high potential for parallelization. Regarding the time-per-slide analysis (Fig. 4C), our tissue detection tool is highly effective (analysis under 10x magnification), with an average time per slide of <1 s (all tests using a standard PC with NVIDIA RTX 3090 24 GB without inference parallelization). Additionally, the artifact detection tool, regardless of the specimen type, allows analysis of single slides in <1 min, with the 5x and 7x versions performing quicker compared to the 10x counterpart. The differences between biopsy and resection cases are not prominent, as all biopsies analyzed contained multiple tissue levels.

**Analysis of misclassifications**
An extensive analysis of the artifact algorithm's accuracy was conducted, with a summary of misclassification patterns provided in Fig. 5. Relevant misclassifications were very infrequent. For example, missing very small artifacts (e.g., very small folds) was sometimes evident, but these might not influence the performance of downstream algorithms applied to the image. An important false positive misclassification occurred in malignant melanoma cases, where the presence of melanin pigment closely resembled air bubble artifacts. This problem was identified early during validation, and additional cases with pigmented melanoma were included in the training dataset. However, in some special cases (e.g., highly pigmented uveal melanoma, example in Supplementary Fig. 7), this misclassification might still occur. A simple solution is to ignore the air bubble artifact class, which is possible due to the multi-class nature of artifact detection. All other misclassifications were either not relevant or of questionable relevance to the downstream analysis and were related to slightly different perceptions of boundaries, imprecise annotations, and overdetection immediately near the true positive artificially changed areas. The most notable inter-artifact misclassification was dark spot/foreign object artifacts detected as air bubble/edge, which is specific to the test dataset (type of synthetic threads used for creating foreign objects) and rarely manifests in real-world implementation (Fig. 5B).

**Evaluation of the algorithm in clinical context: benchmark for pathology institutes and scanning systems**
First, GrandQC can serve as an efficient benchmarking tool for histological slide quality in pathology departments. We performed an analysis

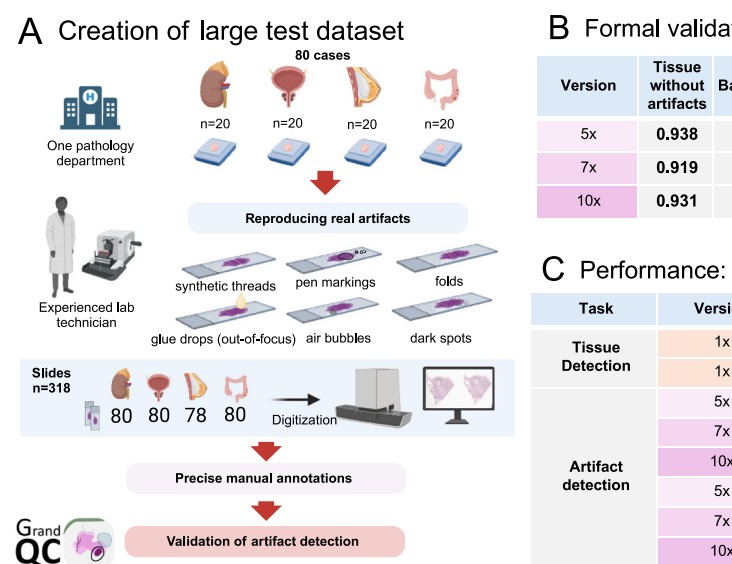

**A** Creation of large test dataset

**B** Formal validation results (segmentation quality: Dice score)

| Version | Tissue without artifacts | Background | Fold | Dark spot & Foreign | Pen markings | Air bubble & Edge | OoFocus | Average Dice |
|---|---|---|---|---|---|---|---|---|
| 5x | **0.938** | 0.732 | 0.786 | 0.406 | 0.966 | 0.846 | 0.820 | 0.785 |
| 7x | **0.919** | 0.756 | 0.801 | 0.325 | 0.979 | 0.856 | 0.863 | 0.808 |
| 10x | **0.931** | 0.761 | 0.828 | 0.528 | 0.981 | 0.890 | 0.846 | 0.824 |

**C** Performance: Time-per-slide analysis

| Task | Version | Slide Type | Min (s) | Max (s) | Mean (s) |
|---|---|---|---|---|---|
| Tissue Detection | 1x | Resections | 0.28 | 1.20 | 0.45 |
| | 1x | Biopsies* | 0.09 | 1.05 | 0.32 |
| Artifact detection | 5x | Resections | 18 | 48 | 27 |
| | 7x | Resections | 18 | 54 | 31 |
| | 10x | Resections | 24 | 72 | 45 |
| | 5x | Biopsies* | 6 | 66 | 29 |
| | 7x | Biopsies* | 6 | 72 | 25 |
| | 10x | Biopsies* | 6 | 102 | 33 |

*Multiple levels in one slide (n=2)

**Fig. 4 | Creation of large test dataset with artifacts and formal validation of artifact detection module. A** Creation of a large test dataset. A large dataset (slides *n* = 318) of slides with artifacts was generated. Representative tissue samples from four organs were taken from one department. Experienced lab technician represented different types of artifacts. These were digitized with a scanner and precisely manually annotated to be used for formal validation of GrandQC resulting in a dataset of 51283, 26571, and 17145 single image patches for 10x, 7x, and 5x extraction magnification, respectively. **B** Formal validation results for artifact detection tasks. All metrics represent Dice scores for segmentation accuracy. Shown are the results for three different algorithm versions (5x, 7x, and 10x). Please note, that even if Dice score for Dark Spot & Foreign is relatively low, this is due to

the fact that there was inter-artifact misclassification. The accuracy of tissue detection without artifact is the most important score (0.919–0.938 dependent on version). Abbreviation: OoFocus−Out-of-focus. **C** Performance analysis of the algorithm: speed of single slide analysis. The metrics are shown for both tissue detection and artifact detection modules, for two different datasets: resection specimens and biopsy specimens (Supplementary Methods; Biopsy specimens with at least 3 levels of tissue per slide). The times provided are only for the step of algorithm processing of all slide patches. E.g., generation of overlay, saving the images as files or any further manipulations which can take certain time are excluded. The test was performed using a typical PC station with a consumer-level GPU card (NVIDIA RTX 3090). Created in https://BioRender.com.

of slides from 19 different pathology departments, representing international and inter-institutional variations in slide and digitization quality. The analysis on the single slide level revealed substantial variations within and among departments. Thus, per slide artifact content (area of artificially changed tissue, any artifacts) showed a variation magnitude of up to 20% when single slides from a department were concerned (Fig. 6A). Based on the slide average, we can stratify all departments according to the quality of WSIs (Fig. 6A). This represents a very useful blueprint for pathology departments that would like to estimate their slide quality and decide if any changes are necessary. Interestingly, P11, demonstrating the best quality among all departments, is a research dataset (PESO dataset for prostate cancer[14]) that was meticulously prepared to allow further registration with immunohistochemistry and therefore not representative of real-world practice. The same analysis concerning different artifact types (Fig. 6B) provides more detailed information as different classes of artifacts are characteristic of different QC situations. For example, tissue folds are a direct correlate for cutting quality (and speed of cutting) and air bubble/slide edge artifacts for slide mounting, while OOF is more complex and might be related to tissue section thickness and scanning system performance (see below). Most dark spot artifacts and pen markings are resolvable and related to suboptimal slide preparation for scanning.

Second, GrandQC is a powerful tool for comparing scanning systems. In Fig. 6C, we compare two of the most popular scanning systems (blinded) using a multi-organ real-world set of histological slides from one department (P19) and show that one scanning system produced much more out-of-focus regions, while the other provided more robust results. Pathology departments can consider this as a benchmark while selecting scanning systems.

Third, GrandQC can be used as a continuous temporal monitoring tool. In Fig. 6D, we show yearly changes in slide quality in one department (P19).

We provide a detailed analysis of WSI quality for one open-source multi-tumor cohort of patient cases (TCGA cohort), being the most popular cohort for research projects in computational pathology, showing high levels of artificial changes in a substantial number of slides and substantial inter-organ variations (Fig. 7).

### GrandQC improves the performance of algorithms in downstream applications

Three use cases (situations) were evaluated to demonstrate how GrandQC can enhance performance of downstream algorithms: (1) for diagnostic multi-class tissue segmentation algorithms: preventing false positive tumor classifications in benign tissue regions, (2) for diagnostic multi-class tissue segmentation algorithms: for improving the segmentation accuracy in tumor regions, (3) for single cell detection/classification algorithms: preventing false cell detections and classifications.

For the first and second use cases, lung[15] and colorectal[4] AI tools were applied. Inn 33 (colorectal) and 105 (lung) ROIs with benign tissue, false positive tumor misclassification were observed in 7 (21.2%) colorectal and 18 (18.1%) lung ROIs due to artifacts. However, none of these regions showed this issue when GrandQC was used to detect and mask the artifacts (Fig. 8A, Supplementary Fig. 8; for dataset details see Methods).

In the second use case, 126 lung and 121 colorectal precisely manually annotated tumor regions were analyzed by diagnostic AI tools before and after GrandQC implementation. OOF regions were synthetically generated in a part of each ROI (see Methods). After detecting and masking artifacts with GrandQC, segmentation accuracy and sensitivity/specificity for tumors and tumor-associated classes (e.g., tumor stroma, necrosis, tertiary lymphoid structures, and mucin) improved significantly (Fig. 8B; Supplementary Figs. 9, 10).

For the third use case, a single-cell detection/classification algorithm was employed to detect six cell types (epithelial/tumor cells and

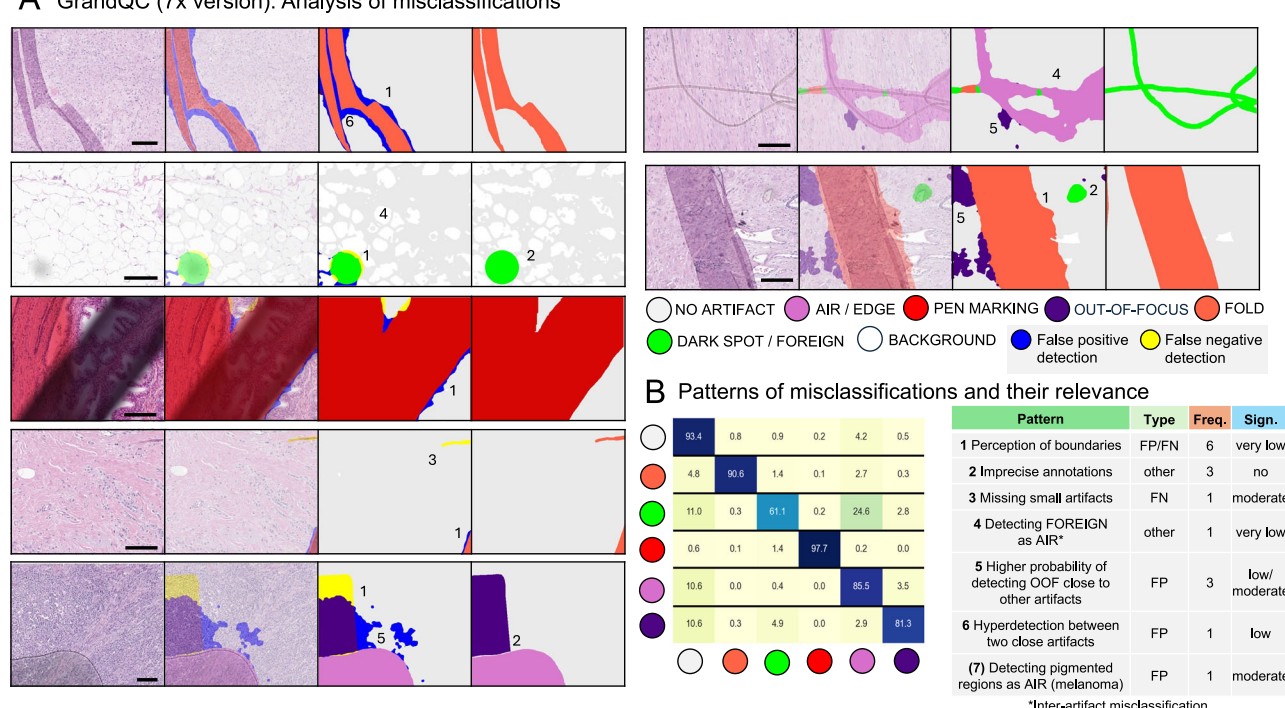

**A** GrandQC (7x version): Analysis of misclassifications

Legend:
○ NO ARTIFACT ● AIR / EDGE ● PEN MARKING ● OUT-OF-FOCUS ● FOLD
● DARK SPOT / FOREIGN ○ BACKGROUND ● False positive detection ● False negative detection

**B** Patterns of misclassifications and their relevance

| | | | | | |
|---|---|---|---|---|---|
| 93.4 | 0.8 | 0.9 | 0.2 | 4.2 | 0.5 |
| 4.8 | 90.6 | 1.4 | 0.1 | 2.7 | 0.3 |
| 11.0 | 0.3 | 61.1 | 0.2 | 24.6 | 2.8 |
| 0.6 | 0.1 | 1.4 | 97.7 | 0.2 | 0.0 |
| 10.6 | 0.0 | 0.4 | 0.0 | 85.5 | 3.5 |
| 10.6 | 0.3 | 4.9 | 0.0 | 2.9 | 81.3 |

| Pattern | Type | Freq. | Sign. |
|---|---|---|---|
| 1 Perception of boundaries | FP/FN | 6 | very low |
| 2 Imprecise annotations | other | 3 | no |
| 3 Missing small artifacts | FN | 1 | moderate |
| 4 Detecting FOREIGN as AIR* | other | 1 | very low |
| 5 Higher probability of detecting OOF close to other artifacts | FP | 3 | low/moderate |
| 6 Hyperdetection between two close artifacts | FP | 1 | low |
| (7) Detecting pigmented regions as AIR (melanoma) | FP | 1 | moderate |

*Inter-artifact misclassification

**Fig. 5 | Analysis of misclassifications in artifact detection. A** The detailed analysis of misclassifications was performed for artifact detection module. The representative examples are shown and summarized in the table in (**B**). The numbers refer to the type of misclassification in the table. Color legends depicts different artifacts and types of misclassifications (false positive artifact detection or false negative artifact underdetection). All scale bars are 100 μm. **B** Patterns of misclassification and their relevance. Left side: the patterns of misclassification concerning class are shown – always horizontally for single color-coded artifacts on the left side. Most prominent inter-artifact misclassification is for DARK SPOT / FOREIGN (green) with 24.6% of detected artifact area misclassified as AIR/EDGE. This is specific to the test dataset used (syntethically generated real artifacts) as for foreign body imitation the synthetic threads were used which are highly reminiscent of the borders of AIR bubble. In the real-world application this misclassification is mostly not seen. Right side: The provided table summarizes the patterns of misclassification. Most misclassifications result from minimal pixel-level variations in perception of object boundaries (1), imprecise annotations (2), or inter-artifact misclassifications (4) and therefore non-significant. Patterns 5 and 6 are also highly tolerable as they extend the area of properly detected artifacts, with 6 being even beneficial. Several misclassification patterns might be relevant to consider for end users and appear in a very limited number of slides: detecting pigmented regions as AIR in malignant melanoma (Supplementary Fig. 3; was addressed partially in course of algorithm development) and overcalling of DARK SPOT/FOREIGN in fatty tissue (pattern 5) in single slides scanned by the 3DHISTECH scanners. These problems can be easily overcome as GrandQC detects single artifacts as different classes, whereby misdetected classes might be specifically ignored. Abbreviations: FP false positive, FN false negative, Freq Frequency, Sign Significance.

five types of immune/stromal cells). The evaluation was conducted by one experienced pathologist (YT) on 200 whole-slide images (WSIs) from patients with colorectal cancer. Results showed that single-cell misclassifications were directly linked to artificially altered regions and could be effectively prevented using the GrandQC quality control step and artifact masking (Fig. 8C).

## Comparison with other QC tools
We conducted an extensive comparison of GrandQC with three open-source quality control tools: HistoQC[6], PathProfiler[7], and HistoROI[8]. Using our large multi-organ test dataset and typical segmentation metrics (Dice score), GrandQC demonstrated substantial superiority over all other tools (detailed analysis in Supplementary Tables 4-6). These findings were also visually confirmed during WSI analysis (examples in Supplementary Fig. 11-25).

## Discussion
Clinical/anatomical pathology as a medical discipline has just started a digital transformation. One of the promises of digital pathology is providing potent AI-based tools to pathologists that span different areas, such as purely diagnostic algorithms and algorithms for more advanced applications, such as prognosis, prediction, and deciphering molecular-genetic alterations from tumor images[1,2]. The pre-analytical step in digital pathology is a major bottleneck for the implementation of downstream AI algorithms, as all histological slides contain different artifacts related to tissue processing and digitization (Fig. 1A) that often lead to the failure of algorithms without any notification. Our own group, in a landmark study[16] showed that any amount and any severity of artifacts can lead to critical misclassifications, such as in tumor detection (one of the most common diagnostic tasks for algorithms). All artifacts can produce false positive and false negative tumor misclassifications[16].

In this study, we developed and extensively validated a powerful tool for quality control in digital pathology (GrandQC). This tool includes two deep learning, pixel-wise segmentation algorithms for tissue detection and multi-class artifact detection, developed using high-quality, large, manually annotated, multi-organ, multi-institutional datasets (Figs. 1B, 2A; Fig. 3A for examples of WSI processing). We open-sourced GrandQC along with a large test dataset with expert annotations that can be used for testing and further development.

The GrandQC tissue detection algorithm performs very well even in the most complicated situations (Fig. 2B), shows high pixel-wise precision (Dice score 0.957), and was additionally validated in a real-world setting using materials from five departments scanned by five different scanners (Fig. 2D). None of the 600 slides assessed by two human analysts showed significant tissue underdetection, with an overall quality score of 9.44 out of 10 points. Moreover, this algorithm is very quick (typical analysis time <1 s for a slide scanned at 400x magnification; Fig. 4C).

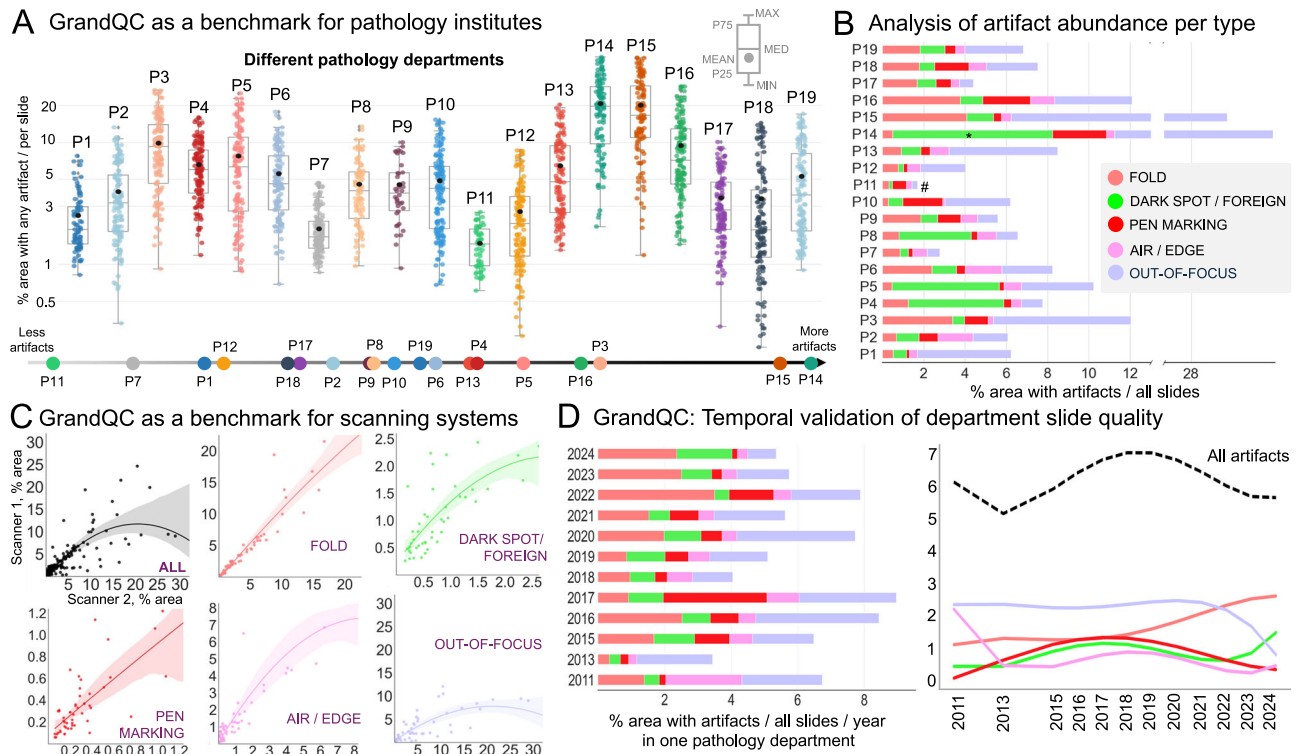

**Fig. 6 | Investigating GrandQC as a benchmark for pathology institutes and scanning systems. A** GrandQC effectively benchmarks slide quality by allowing pathology institute to assess artifact frequency and area in sample slides. We analyze slides from 19 different pathology departments: point plot represents % area for all artifact types in single image. All departments are placed on the line dependent on the mean % area of artifacts in the whole slide set serving as a benchline and reference for new departments assessing slide quality. In the box plots, center line represents the median, black points show the mean, box bounds indicate interquartile range (IQR), covering the 25th to 75th percentiles. Whiskers extend 1.5 times the IQR, gray points denote outliers beyond this range. **B** Detailed analysis for pathology departments concerning single artifact type (% area occupied by artifacts in the whole dataset). Comments: *In P14 pathology department (3D HISTECH scanner) we observed overcalling of properly detected DARK SPOT (attributed to dust particles and greasy fingerprints). # is a prostate dataset prepared by the pathology department for a project on immunohistochemistry

registration (PESO dataset). This is exceptionally high quality, uncommon for regular pathology departments which should be considered in research projects. **C** GrandQC as benchmark for scanning systems. Two most common scanning systems were tested on a heterogeneous dataset from University Hospital Cologne, containing different organs/specimens (each slide with two scanners). Single points represent slides with % area occupied by all or single artifact being coordinates: x–Scanner 1, y–Scanner 2. 95% confidence interval was used. While artifact areas showed general concordance. Scanner 2 produced significantly more out-of-focus regions, making it a notable benchmark for departments selecting scanning systems to optimize slide quality. **D** Temporal validation of slide quality serves as a third benchmark, enabling pathology departments to monitor changes over time. Data from University Hospital Cologne is shown on a yearly basis. Analysis can also be done daily, weekly, or monthly, with added outlier detection. Analyzing different artifact types reflects various aspects of slide preparation, from lab work to digitization. Source data is provided as a Source Data file.

Several algorithms have been published for tissue detection, such as tissueloc[17], HistoQC[6], or TIA Toolbox[18]. Most of these tools utilize pixel value analysis, such as simple threshold and Otsu thresholding[19] using grayscale-transformed WSIs, among others[20]. These methods are simple but have many known limitations related to tissue detection with increased transparency or tissue/background intermingling (e.g., fatty tissue, breast tissue, lung tissue), overstained slide backgrounds, understained slides with very faint staining, contamination in the slide, and overdetection of numerous artifacts such as air bubbles, glass edges, and dark spots (scratches) as tissue. Bandi et al.[20] compared non-deep-learning methods (foreground extraction from structure information) with a fully convolutional neural network and U-Net based approach, showing significant superiority of deep-learning-based approaches (Jaccard Index Mean 0.937 vs 0.870). Therefore, algorithmic non-AI methods cannot be used reliably for clinical-grade tools, whereas deep learning-based algorithms can provide more accuracy and reliability. To the best of our knowledge, the GrandQC tissue detection module is the first high-precision tissue detection algorithm to be open-sourced for free usage.

As for the artifact detection module (Figs. 2A, 3), we developed and extensively validated three different versions (for 5x, 7x, and 10x magnification) that offer an excellent trade-off between precision and

inference speed. All three versions allow detection of non-artificially changed tissue with high segmentation accuracy (Dice scores 0.919–0.931; Fig. 4B). For validation purposes, we created a large multi-organ test dataset (Fig. 4A). We engaged an experienced lab technician to reproduce all artifacts, resulting in 318 slides that were annotated by experienced human analysts, leading to a large test dataset comparable in size to the training dataset used.

Moreover, we performed a detailed analysis of misclassifications, showing that most misclassifications at a pixel level are not relevant for downstream analysis and are related to slight deviations in perception of object boundaries, imprecise annotations, a higher probability of detecting artifacts close to other artifacts (which leads, for example, to the filling of small spaces between parts of tissue folds), and some inter-artifact misclassifications (Fig. 5). Two relevant misclassification patterns affected a small number of slides: (1) Detection of pigment in malignant melanomas as an air bubble. This problem, which arose during algorithm testing, initially led to a second round of development, whereby we included slides with malignant melanoma into the training dataset and retrained the algorithm. Nevertheless, this problem manifested in a few malignant melanomas with very high levels of pigmentation (uveal malignant melanoma, Fig. 7: TCGA dataset, UVM cohort). As our artifact detection algorithm is multi-class, the obvious

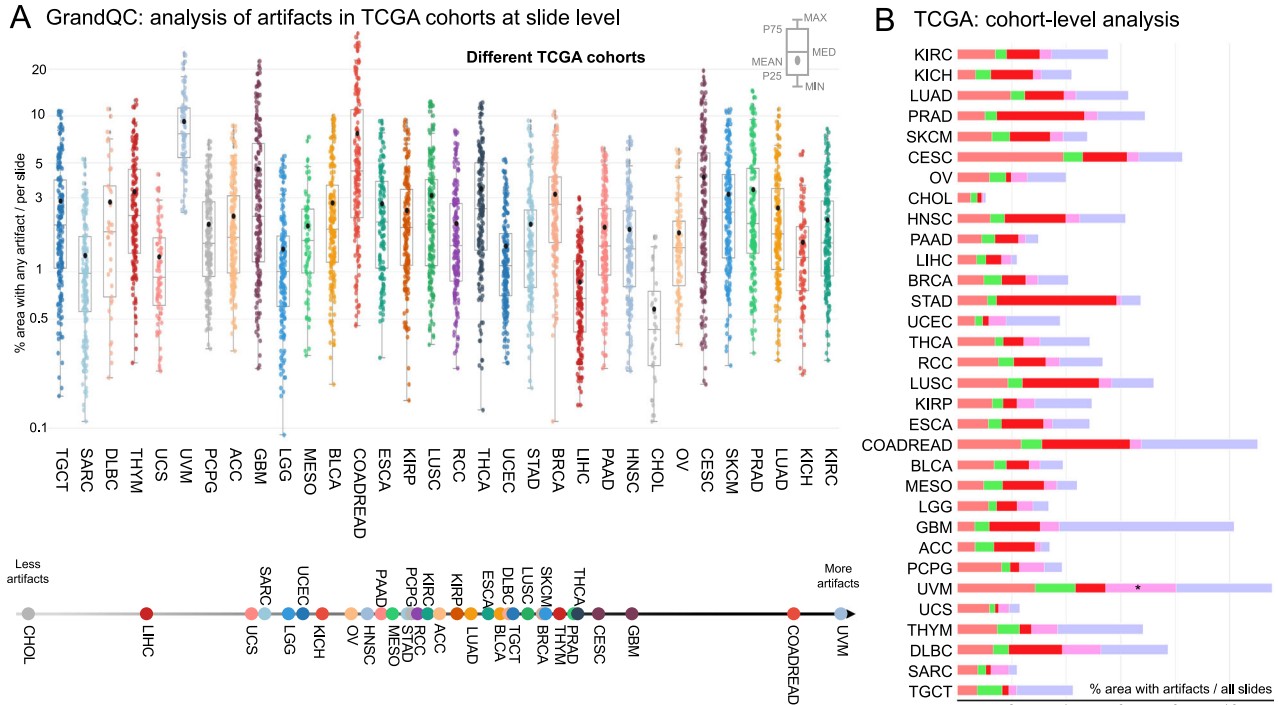

**Fig. 7 | Analysis of artifacts in the largest open-source multi-organ malignant tumor cohort (TCGA) used by research groups worldwide. A**, **B** The same analysis as in Fig. 6 is provided for the largest open-source research cohort (The Cancer Genome Atlas) containing multiple cohorts of patient cases and slides to different types of malignant tumors. Only diagnostic slides were analyzed (formalin-fixed paraffin embedded tissue). **A** Slide level analysis of % area occupied by all artifacts (above) and stratification of cohorts (below) based on the mean value. **B** Cohort level analysis for single types of artifacts. Comments: * Uveal melanoma (highly pigmented) cohort with focal misclassification of pigmented areas as AIR (high similarity). Refer to https://gdac.broadinstitute.org/ for abbreviations of cohort names. All artifacts and tissue detection masks for all TCGA cohorts are open-

sourced for academic research purposes. This is of utmost value for research projects with non-supervised approaches. Using the provided masks allows to remove artificially changed areas from training ($n$ = 2–10% of slide area) which might be an important confounder for algorithm biases and inaccuracy. For the box plots in Figure (**A**), the center line represents the median, the black points represent the mean of the value, the box bounds depict the interquartile range (IQR), covering data from the 25th to 75th percentiles, which represents the middle 50% of the values, the whiskers extend to a range of 1.5 times the IQR from the lower and upper quartiles, capturing a broader spread of the data, and the gray points are the outliers beyond the whiskers. Source data is provided as a Source Data file.

solution is to ignore air bubble artifacts in such cases. (2) Substantial overdetection of dark spot artifacts (mostly fingerprints) in the fatty tissue of some slides in one cohort (Fig. 6: Institute P14) scanned by the 3DHISTECH scanning system. This was, however, not the case in another cohort scanned by the 3DHISTECH scanner (Fig. 6: Institute P15), so we consider this to be an institute-specific situation.

Several QC tools were developed and published earlier[6–13,21–23]. The HistoQC tool[6] is a tool that utilizes non-deep-learning methods based on pixel analysis. It allows for detection of air bubble, slide edge, out-of-focus, and pen markings artifacts. Although HistoQC is a powerful tool for analysis of domain shifts within a cohort of cases (such as staining, color characteristics), it was not formally validated at the pixel-level for artifact detection by developers, only at the slide-level against human analysts[23] (disqualified/qualified WSIs for only one category of cases: kidney biopsies). A recent analysis by Rodrigues et al.[24] showed an F1-score/Dice score of 0.54 for artifact detection (dataset: 49 annotated WSIs). GrandQC showed Dice-score of 0.919–0.938 (dependent on version) for artifacts vs. non-artificially changed tissue, which is superior by a large margin. Similarly, Haghighat et al.[7] showed an AUC of 0.67 for HistoQC in detecting the tissue without artifacts. This supports the fact that deep learning allows building substantially more powerful tools and is a state-of-the-art technology for the segmentation of complex objects in histological slides, including artifacts[1,2]. Smit et al.[13] built a DL-based, multi-class artifact detection tool (DeepLabV3 + /EfficientNetB2) using a training dataset of 142 images, 21 of which were used for internal validation. Although the authors show competitive pixel-level accuracies (0.70–

0.97), there were no external validation. In our study, we used a large external test dataset of 318 images with extensive manual annotations. Moreover, Smit et al.[13] do not open-source the model and provide access only via a Grand Challenge platform, implying necessity of uploading the slides, which is not practicable under real-world conditions and might interfere with privacy regulation. Patil et al.[8] train a classification model, HistoROI (patch-level, does not allow pixel-wise segmentation which should be considered a state-of-the-art for this application) for breast cancer that includes artifacts as a class. Interestingly, when validated on CRC-100k dataset[25], 97% of background patches from this dataset were detected as the artifact class by the model. Hemattirad et al.[22] used a YOLO-v4 model for object detection and trained it using 92 mapped WSIs. It was validated only internally on 15 WSIs. Importantly, YOLO produces bounding boxes around the objects that include large areas of non-artificially changed tissue, depending on the form of the artifacts. This is not desirable, with pixel-wise segmentation being a more effective solution. The model is not open-sourced. Haghighat et al.[7] developed PathProfiler using only one histological domain (prostate specimens) which is a patch-level classification algorithm (ResNet18; analyzing WSI in coarse regions) using an interesting scoring system for defining slide usability. It cumulates scores from several parameters, including staining quality and the presence of OOF and folding artifacts. Although this tool was open-sourced, its application might be limited to prostate domain, as shown by the authors in a subsequent validation study[21]. Moreover, our tool has several significant advantages: multi-organ nature, precise pixel-wise segmentation instead of patch-level classification, and detection

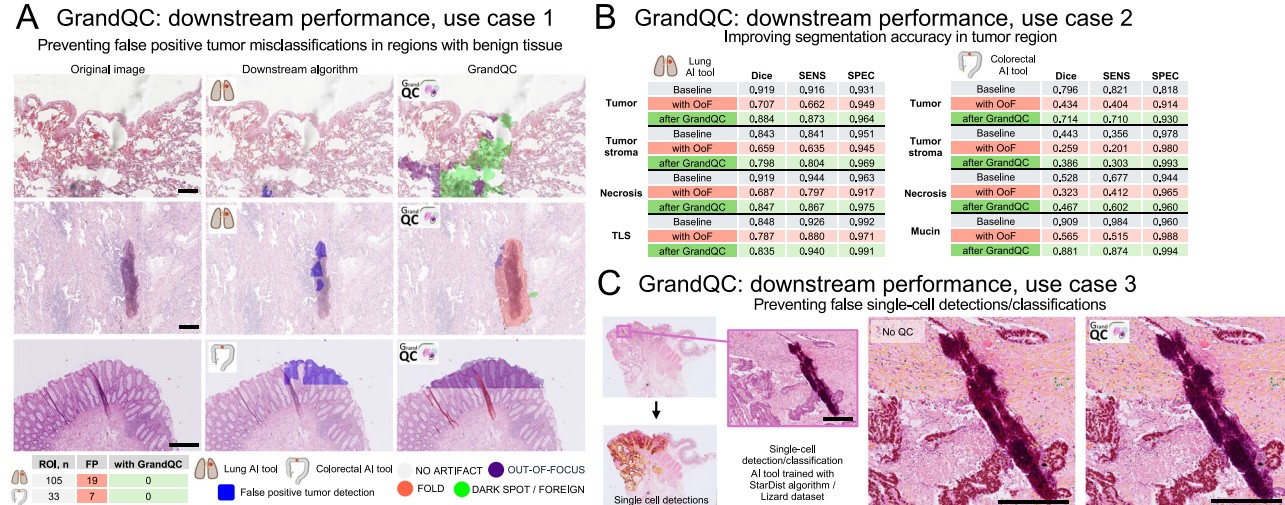

**Fig. 8 | GrandQC improves performance of diagnostic algorithms in downstream applications.** Three use cases (situations) were evaluated to demonstrate the ability of GrandQC to improve the performance of downstream algorithms: **A** For diagnostic multi-class tissue segmentation algorithms: preventing false positive tumor classifications in benign tissue regions. Representative examples demonstrate how the detection and masking of artifacts help prevent false positive misclassifications in regions of interests containing benign tissue from two diagnostic domains (colorectal, lung). These regions were analyzed using previously developed multi-class tissue segmentation algorithms for lung and colorectal cancer, respectively. **B** For diagnostic multi-class tissue segmentation algorithms: for improving the segmentation accuracy in tumor regions. A total of 126 regions of interest (ROI) were analyzed for lung cancer and 121 for colorectal cancer, resulting in 2,016 and 1,936 patches, respectively, each sized 512 × 512 px (MPP 1.0). In these ROIs, synthetic out-of-focus regions were generated. The segmentation accuracy results are shown for each of the diagnostic tools (lung and colorectal), comparing a baseline (ROIs without artifacts), OOF regions without quality control, and OOF regions with artifact detection and masking using GrandQC. Notably, while GrandQC detects and masks artifacts, preventing misclassifications, it does not enhance the algorithm's ability to detect structures in the affected areas. Therefore, thresholds may be needed to prompt pathologists to conduct additional reviews when large artifact areas are detected in certain context, as these may obscure important findings. **C** For single-cell detection/classification algorithms: preventing false cell detections and classifications. An algorithm was trained for single-cell detection (epithelial/tumor cells and five classes of immune/stromal cells) and classification in colorectal cancer. These algorithms are particularly vulnerable to artifacts due to the small size and subtle features of the cells being analyzed. The case presented is shows how a tissue fold artifact led to false cell detections and their misclassification as epithelial/tumor cells (brown; various other colors represent other cell classes). This issue can be easily prevented through a pre-analytical step using GrandQC, which masks artifacts before downstream processing. All scale bars are 200 μm.

of a larger number of artifacts. Several other studies addressed the problem of OOF[9,11,12] or tissue fold[10] artifacts, but not in context of a comprehensive multi-class QC tool. In our study, we perform direct comparison of GrandQC to three of the above mentioned tools (HistoROI, PathProfiler, HistoQC) showing that our instrument outperforms these tools by a large margin (Supplementary Tables 3-6, Supplementary Fig. 11-25).

During our study, we take the next step to test our tool as a benchmark for histological slide quality in different pathology departments. In an extensive validation, we analyze routine sections from 19 pathology departments digitized by different scanning systems, showing high variations among institutions concerning overall artifact content (Fig. 6A) and single artifact types (Fig. 6B). We compare and rank the departments according to quality (Fig. 6A). Therefore, the GrandQC tool can be used as a benchmark: each new pathology department wishing to evaluate its quality can use GrandQC and see where it is located on the slide quality line. Moreover, GrandQC can be used for historical (inter-year comparison; Fig. 6D) or real-time (e.g., on a daily or weekly basis) monitoring of slide quality to detect QC issues early that should be corrected. Additionally, GrandQC can be effectively used for the comparison of scanning systems, especially when making decisions about digitization. Scanning systems might perform with substantial differences in a specific quality of a pathology institute, and GrandQC allows the selection of the optimally performing scanning system for a particular institute (Fig. 6C). Lastly, we analyze the diagnostic slides of all TCGA cohorts, being the major research resource for computational pathology projects worldwide, compare their quality (Fig. 7), and notably, open-source all QC masks that can be used to exclude artificially changed areas during training (especially weakly or self-supervised learning) or inference.

In our study, we provide evidence that employing GrandQC can significantly improve the performance of downstream algorithms (Fig. 8) clearly demonstrating the importance this preanalytical step for application of AI tools in pathology. This, together with open-sourcing GrandQC itself and a large, manually annotated test dataset, is a major contribution to the computational pathology field, capable of solving the QC problem.

Our tool is not devoid of limitations. Although it performs very well with QC of H&E-stained images, it was not trained to work with immunohistochemistry slides, which is an area for further development. Our tool is not intended for the analysis of domain shift issues such as staining variability, where several excellent tools are already available (e.g., HistoQC). In our study, we chose not to use state-of-the-art foundation encoders like UNI, Prov-Gigapath, or CONCH[26–28] for model development due to their current limitations in applying to multi-class semantic tissue segmentation. These limitations include small patch sizes, training at high magnifications (20-40x), and computational inefficiency, among others. For our validation of tissue detection in clinical context, we used a semi-quantitative scale which is subjective and cannot be used for comparison of different methodologies. The aim was rather to identify the number of slides that will pass the "clinical" tissue detection QC check.

Drawing a conclusion, we developed a powerful tool (GrandQC) for digital pathology, consisting of tissue and multi-class artifact segmentation modules, allowing quick and precise QC of H&E-stained histological slides. GrandQC can effectively pre-process the slides for any form of diagnostics or analysis by downstream image analysis algorithms. We open-source the tool, a large manually annotated test dataset, and full QC masks for all TCGA cohorts, which is a major contribution to the field. GrandQC is an effective benchmark for

pathology departments and scanning systems, allowing pathology departments to select the optimal scanning system for their particular slide quality while considering digitization.

## Methods

### Ethical aspects

All study steps were performed in accordance with the Declaration of Helsinki. This study was approved by the Ethical committee of the University of Cologne, University of Essen, Medical Faculty of Martin Luther University Halle-Wittenberg, University Hospital Leipzig, Charité (joint 22-1233, Project FED-PATH; joint Cologne/Charité 20-1583), the Ethical committee of Lower Austria (GS1-EK-4/694-2021), Kameda Hospital (22-094), University Hospital Aachen (EK 405/2). The need for patient consent was waived as only anonymized, retrospective materials were used.

### Training datasets

A dataset comprising 208 whole-slide images (WSI) for the tissue detection task and 420 WSI (Hematoxylin & Eosin, H&E, stained) for the artifact detection task (with partial overlap) was created. This heterogeneous dataset was constructed from WSIs from three main sources representative of a broad spectrum of institutes, lab techniques, and different types of organs and specimens (biopsy, resection specimens). The details are outlined in Fig. 1B. The WSIs from the training dataset were digitized using different models of Leica (GT450, AT2, CS2) and Hamamatsu scanners (Nanozoomer S360, S60), mostly at 40x magnification (micron/pixel, MPP, ~0.25).

### Annotations

For the tissue detection task, the annotations were automated in QuPath v.0.4.3 software[29] (Supplementary Methods). For the artifact detection task, high-quality, precise annotations were generated manually representing eight classes (including non-artificially changed tissue and seven types of artifacts, Fig. 1C). Background (as a separate class) was generated using QuPath Thresholder (Supplementary Methods). Dense annotations were created in most regions. The exceptions involved regions with numerous small artifacts (typically dark spots), where the surrounding areas are a mix of small out-of-focus regions and unaffected regions, where it is difficult to establish a ground truth. In these cases, we applied sparse annotations to mark the artifacts while leaving the surrounding areas unannotated.

### Generation of synthetic OOF

Although out-of-focus (OOF) areas were annotated, this is challenging due to uneven borders and spatial association with other types of artifacts (statistical bias). OOF regions were generated synthetically to enrich the training dataset (Supplementary Fig. 1; Supplementary Methods) and balance different levels of OOF severity, as well as amplify the number of regions where OOF directly contacts tissue without artifacts.

### Test dataset

Using the original approach, a comprehensive test dataset was generated using 318 histological slides from 80 cases (4 organs/tumor types) from one pathology department (Wiener Neustadt, Austria; Fig. 4A). An experienced pathology lab technician reproduced real artifacts using different methods: creating folds through light mechanical traction, contaminating sections with foreign objects (e.g., threads), creating air bubbles through inappropriate cover glass placement, out-of-focus areas via glue drops on the cover glass, dark spots via fingerprints or dust, and pen markings on a subset of slides. These slides were digitized using a Leica GT450 scanner at 40x magnification and extensively annotated by a human analyst, resulting in a test dataset with an annotated area comparable to the training dataset. The OOF regions were manually annotated (without synthetic

generation). This dataset was used for the formal validation of the developed tissue and artifact detection algorithm.

### Algorithm development

Both algorithms (tissue detection and artifact detection) are pixel-wise semantic segmentation methods (for details, see Fig. 2A). Considered as hyperparameters, different encoder and decoder architectures were tested. For details to algorithm development, architecture and hyperparameter choices see Supplementary Table 1. The final architecture for both algorithms included EfficientNetB0 as the encoder and UNet + + as the decoder. For full details on training procedures and hardware used see Supplementary Methods.

### Performance evaluation of the algorithm

The abovementioned test dataset was used for formal validation based on pixel-level segmentation accuracy metrics (Dice score). Moreover, human experts were involved in validating the tissue detection algorithm in a real-world setting. For this purpose, an additional dataset comprising 600 slides without pre-selection (Fig. 2D) from each of five different institutions (five different scanning systems) representing real-world practice was constructed and evaluated by two experts.

### Dataset for runtime performance analysis

One hundred cases from one department were used for runtime performance analysis (radical prostatectomy or prostate biopsy specimens, Supplementary Fig. 2): resection cases ($n = 50$; one tissue level per slide) and biopsy cases ($n = 50$; 2-3 tissue levels per slide). These slides were scanned by a Hamamatsu S360 scanner at 400x magnification.

### QC benchmark for pathology institutes and scanning systems

The developed tool for artifact detection was implemented as a quality control benchmark for pathology institutes, allowing tracking and quantitative assessment of histological slide quality compared to other pathology departments. The evaluation included materials from 19 pathology departments (datasets from previous studies, open-sourced datasets, proprietary datasets). For details on the datasets, see Supplementary Table 2. The benchmark was implemented for single slides, all slides of the institute, slides scanned with different scanning systems, and slides from different years spanning over 13 years of practice. Two scanning systems were used for detailed comparison in a scanning system benchmark (Leica GT450 and Hamamatsu NanoZoomer S360; in all further analyses, the type of scanning system is blinded).

### Evaluation of downstream algorithm performance in context of quality control

Three use cases (situations) were evaluated to demonstrate the ability of GrandQC to improve performance of downstream algorithms: (1) for diagnostic multi-class tissue segmentation algorithms: preventing false positive tumor classifications in benign tissue regions, (2) for diagnostic multi-class tissue segmentation algorithms: for improving the segmentation accuracy in tumor regions, (3) for single cell detection/classification algorithms: preventing false cell detections and classifications.

For first two situations, two previously developed and extensively validated clinical-grade algorithms for lung[15] and colorectal[4] specimens were employed. Both are UNet + +-based semantic segmentation algorithms working with a patch size of 512 × 512 px at MPP 1.0 (roughly 10x objective magnification).

For use case 1, a dataset was generated from benign tissue regions of interest (ROI) of whole-slide images containing artifacts for both lung (WSI/patient $n = 40$, lung adenocarcinoma $n = 20$/squamous cell carcinoma $n = 20$; ROI $n = 105$) and colorectal (WSI/patient $n = 30$; ROI $n = 33$) domain, respectively. For further description of the cases/slides refer to original publications. The ROIs were analyzed by the lung or

colorectal AI tools with and without GrandQC. The number of ROIs with false positive tumor detections due to artifacts was recorded (with and without quality control step).

For use case 2, same WSI datasets were used, however, now the manually annotated tumor regions were extracted and analyzed. These contain tumor and tumor-associated classes (colorectal: tumor stroma, necrotic debris, and mucin; lung: tumor stroma, necrotic debris, and tertiary lymphoid structures). The number of ROIs was 126 for lung and 121 for colorectal WSIs, resulting in 2,016 and 1,936 patches of size 512 × 512 px (MPP 1.0), respectively. As initially the manual annotations were made in areas without artifacts, a synthetic OOF regions were generated. The approach was similar to the one used for training data, where Gaussian blur (with kernel sizes from 3 to 17 with the same distribution as the training dataset; Supplementary Fig. 1) was applied to the original image. This was done using random binary maps to ensure that at least 30% of the ROI was out of focus. The resulting images can be viewed in Supplementary Fig. 9 and 10. The segmentation accuracy as well as sensitivity and specificity was derived from baseline analysis (native ROIs), with OOF without QC step, and with OOF with QC step (artifact detection and masking by GrandQC).

For use case 3, we trained a single-cell detection/classification model using StarDist[30] algorithm and Lizard[31] dataset. Training was performed with StarDist default setup for CoNIC challenge[31]. Single-cell detection/classification was performed on 200 WSIs of colorectal cancer from University Hospital Cologne, before and after QC. The results of single-cell detection/classification in artificially changes areas were evaluated visually.

### Comparison with other QC tools

Three open-source QC tools were used for a direct comparison with GrandQC: HistoQC[6], PathProfiler[7], and HistoROI[8]. These tools were implemented using their default configurations as recommended by the developers. Each QC tool is designed to detect different types of artifacts: HistoQC primarily focuses on identifying usable tissue regions, PathProfiler detects tissue folds, staining errors, and OOF issues, and HistoROI detects all artifacts as a single class. To enable direct comparison, we consolidated detected artifacts into one or more comparable classes (Supplementary Table 4-6).

In the first experiment, we applied each tool to our large, manually annotated test dataset and measured their performance using the Dice score. When comparing GrandQC to HistoROI, we merged GrandQC's artifact detections into a single "Artifacts" class, and used HistoROI's non-artifact detections as the equivalent of GrandQC's "Tissue without artifacts". In the comparison with PathProfiler, we evaluated the "Tissue without artifacts", "Folds," and "Out-of-focus" classes, combining GrandQC's remaining artifact detections into "Other artifacts." For the comparison with HistoQC, we assessed "Tissue without artifacts" and "Artifacts + Background" classes, where HistoQC's usable tissue regions were treated as equivalent to GrandQC's "Tissue without artifacts," and the background was included in the "Artifacts" class for alignment. In the second experiment (visual validation), we considered all types of artifacts as one single class, independent of the tool type.

### Statistics and reproducibility

Given the nature of the study no statistical method was used to pre-determine sample size. No data were excluded from the analyses. The experiments were not randomized. The Investigators were not blinded to allocation during experiments and outcome assessment.

### The use of AI and AI-assisted technologies in scientific writing.

The assistive technologies were used to improve readability and language of the manuscript. This was done under human oversight and control.

### Reporting summary

Further information on research design is available in the Nature Portfolio Reporting Summary linked to this article.

## Data availability

The datasets and models generated or used in this study have been made available with following ways: QC masks generated from the TCGA public dataset are deposited at Zenodo under accession code (https://zenodo.org/records/14041578). The test datasets (image patches with corresponding ground truth maps of manual annotations), stratified by organ are deposited at Zenodo under accession code (https://zenodo.org/records/14039591). The whole-slide images from TCGA cohort are available from https://portal.gdc.cancer.gov/. PESO dataset is available from https://zenodo.org/records/1485967. Wilkinson et al. dataset and Madabhushi et al. datasets are available from The Cancer imaging archive (https://www.cancerimagingarchive.net/). Proprietary data is not being released due to authors institutions' data privacy regulation. All other data is available per request from corresponding author. Source data are provided with this paper.

## Code availability

GrandQC is being open-sourced (https://github.com/cpath-ukk/grandqc) for academic research and non-commercial use by pathology departments only. The code is also deposited at Zenodo under accession code (https://zenodo.org/records/14062356). The use for commercial purposes is not allowed. The trained model checkpoints for three different magnifications are deposited at Zenodo under accession code (https://zenodo.org/records/14041538).

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

## Acknowledgements

This project was primarily funded by Federal Ministry of Education and Research of Germany (BMBF): Project FED-PATH (Y.T., R.B., F.M., C.W., H.B.). Y.T. and Z.W. had access to all the data. The funding organization did not participate in any aspects of study planning, design, implementation, data analysis, or formulating the conclusions of this study. We thank the Regional Computing Center of the University of Cologne (RRZK) for providing computing time on the DFG-funded (Funding number: INST 216/512/1FUGG) High Performance Computing (HPC) system CHEOPS as well as support.

## Author contributions

Z.W.: Data management and preparation, development and validation of algorithms, formal experiments, data analysis and interpretation, manuscript drafting. A.S.: data management, annotations of training dataset, data analysis and interpretation. A.P., F.M., C.W., M.B., L.G., H.B., T.L., D.J., S.S., A.B., J.F., M.B., B.S.M., A.T., G.N.: data preparation and management, validation study. S.K.: data analysis. A.P., W.H.: test dataset preparation, data management. K.B.: planning of experiments, supervision. A.Q., R.B.: data management, resources. Y.T.: conception and design, data annotation (training dataset), algorithm training and evaluation, data analysis and interpretation, manuscript drafting, supervision, resources. All authors: manuscript review and editing, critical revision for important intellectual content.

## Funding

## Competing interests

The authors declare no relevant competing interests.
