## [Transparent Peer Review file · Nature Communications]

GrandQC: a comprehensive solution to quality control problem in digital pathology

Corresponding Author: Mr Zhilong Weng

Version 0:

Reviewer comments:

Reviewer #1

(Remarks to the Author)

In this paper, the authors propose GrandQC, a computer vision algorithm for artifact detection and segmentation algo on pathology whole-slide images (WSIs). The algorithm performs several useful functions, most notably segmenting the tissue in a slide and performing a multi-class artifact detection and segmentation. The algorithm can be considered a base for other digital pathology workflows, and could be used to do things that the authors propose such as assess hospital-specific failure modes or trends over time in digitization.

There are many useful characteristics of this study, most notably:

- release of the model as open-source
- release of prediction masks on TCGA
- the diversity of data sources used to train the model
- strong performance of the model
- careful consideration into how to do the classification of different artifacts

The authors have thought carefully about how to design this tool, and it is well-reflected in the work.

However, although this is a meaningful contribution to the field of applied computer vision in pathology, I do not think this study is of broad interest or novelty to support its acceptance at the current venue. There are several major concerns supporting this conclusion:

1. The work is not novel in its approach, it is largely similar to existing tech in the field. The main novelty is in the open-access nature of the model. This is a very nice novelty, but makes it more suitable for a more specialized venue.
2. The approach is not state-of-the-art, it is more dated than e.g. recent results showing the utility of foundation models for pathology, which use multiple orders of magnitude greater numbers of slides and patches. Again, this doesn't prove troubling for the actual application which is more straightforward and may not necessitate SOTA ML methods.
3. The dataset is not particularly large or diverse in comparison to other published methods for ML in pathology, which may limit generalizability of the model (see e.g. recent results from Faisal Mahmood's lab).

This remains a useful contribution, however, due to the open source nature of the model and the other novelties in the approach.

I have additional questions regarding the approach taken, where some additional detail is necessary to evaluate the results. These are more minor:

- How does performance compare to pathologist performance at the tasks? I.e., do two pathologists agree on artifact areas about as much as the model agrees with each pathologist?
- Why does one need models at both 5x and 10x? If scanned at 10x, couldn't one simply pass the second layer of the WSI pyramid through the 5x model?
- More broadly, is "10x" the appropriate measurement of a length scale here? A more precise length measurement would be the edge length of a pixel from the scan, e.g. 1.0 micron per pixel (mpp).
- What does it mean that dense annotations were created "where possible?" Was there a specific strategy for choosing

where to densely annotate?

- Clinical validation typically refers to evaluation of a tool for a specific purpose with predefined acceptance criteria. The CV shown here is a useful evaluation of the model performance, but does not involve the rigor of a clinical validation (e.g. power calculations, AC, etc.). "Model evaluation" may be a more appropriate term, but the authors should be careful in describing the evaluation work performed.

Reviewer #2

(Remarks to the Author)

Summary

The authors present grandQC, a quality control model for tissue and artifact detection in HE stainings. Their model is developed using a large multi-organ and multi-institutional dataset. They extensively test and validate its performance and show the validity of the model for different applications including benchmarking of staining quality for pathology departments, and assessing staining quality over time. The manuscript is suitable for the journal and will be of significance to the field. The open-sourceness of the model and the annotations will be of significant help for further developments. Having said that, there are a number of points that require further work before the manuscript is ready for publication.

Major comments

- 1- The relevance of artifact removal or artifact misclassification in downstream analyses is mentioned several times in the text. Examples include abstract (lines 49-50), results (lines 257-259), among others. The authors provide several references but there is no demonstration of how grandQC improves downstream analysis algorithms. The manuscript would gain significant impact if this potential improvement is quantified in a couple of examples.
- 2- The authors do not compare the performance of GrandQC with any other method. I agree that other methods might not be directly comparable, but comparisons can be made. They mention a number of these in their manuscript: HistoQC, PathProfiler, HistoROI, etc.). Reporting the results obtained in the corresponding manuscripts might not be relevant as the performance metrics might be dataset dependant. It is necessary that whenever applicable, the authors compare grandQC with other SOTA methods (for the type of artifacts that are applicable).
- 3- In line with the previous point, the authors report the results of their model per artifact type but not per pathology center. Are there significant differences in predictive performance of GrandQC based on pathology department?
- 4- The added value of the tissue detection module in the artifact detection module is unclear. The authors should perform an ablation study (or similar) to check the gain in performance (if any).
- 5- Methods, lines 155-156: the authors mention that the test dataset was used to validate the tissue detection algorithm. Is this correct? Why was this dataset not used to validate the artifact detection algorithm? The dataset was specifically developed for the generation of artifactual areas.
- 6- The authors mention that different encoder and decoder architectures were tested. They also mention that full details on the training procedures in the supplementary methods. However, nothing is mentioned regarding the architectures tested nor results are reported for different hyperparameter choices. Nothing is mentioned regarding the number of epochs, loss functions, or anything of the like. A much more thorough description of the training procedures as well as the obtained results (also for the non-optimal configurations) is required.
- 7- The authors mention that during training, the air bubbles and the slide edge artifacts are merged due to their visual similarity. The same is true for dark spot and foreign object artifacts. They also mention that most misclassification problems happen between artifact types. However, they claim several times in the text that the multiclass nature of their model allows for flexibility. This is particularly noticeable when reporting the issue with pigmented tissue in melanoma slides, saying that the air bubble class could be disregarded (results, lines 264-265). Those two aspects seem a bit contradictory: sometimes the multiclass segmentation nature of the model is emphasized highlighting the relevance of getting class-level predictions; other times, they say that having the misclassifications between artifact types is not relevant. The manuscript would benefit from coherence in this matter.
- 8- Results, lines 218-223: The authors present a custom score for tissue detection quality based on subjective visual evaluation by two experienced analysts. This custom score's value is limited without comparison to other methodologies in a blind setting.

Minor comments

- 1- Abstract, lines 63-64: the authors mention that the segmentation of artifacts and tissue without artifacts (dice score 0.919-0.938). This is misleading as those values are only for the areas without artifacts. The reported dice scores for the artifacts themselves in Figure 4.B is significantly lower (0.325-0.528 for Dark spot & foreign). The result is reported in the same way in the results (lines 237-238).
- 2- The authors mention that the annotations were performed manually. However, in the methods, they describe that the tissue segmentation annotations were automated in QuPath. Please correct.
- 3- The authors provide three models at 5X, 7X, and 10X magnification. Since the slides were scanned at 40X magnification, each pixel at 5X represents 64 pixels at 40X. This discrepancy might explain issues with detecting small artifacts. It raises the question of whether training at higher magnification could improve artifact detection in those cases. Downscaling the images could also impact the detection of out-of-focus (OOF) artifacts. It would be nice to check if this is the case.
- 4- Methods, lines 185-186: the reference regarding supplementary figure 1 appears to be incorrect. The authors are referring to the details on the datasets but supplementary figure 1 refers to the artificial generation of OOF areas.
- 5- Results, line 230: The authors refer to supplementary figures 4 and 5 for the comparative of the different versions. However, the correct figures seem to be supplementary figures 3 and 4. Additionally, these figures do not show the ground truth and the provided magnification of the figures makes their interpretation challenging. Moreover, at least with the provided example, there seems to be quite some disagreement between all three models. A concordance study should be conducted to evaluate the differences in predicted areas between the models.

6- Typographic error in line 119: AA.

Reviewer #3

(Remarks to the Author)

I co-reviewed this manuscript with one of the reviewers who provided the listed reports. This is part of the Nature Communications initiative to facilitate training in peer review and to provide appropriate recognition for Early Career Researchers who co-review manuscripts.”

Version 1:

Reviewer comments:

Reviewer #1

(Remarks to the Author)

I am grateful to the authors for their response to my review. I will respond a bit below:

1. It is very helpful to compare this method to other approaches, however, this does also highlight that the modeling approach is not radical, rather a notable step forward from prior approaches. Again, the key contribution of a tool that is open-source, performant, and compute-efficient is meaningful, but moreso within the field of computational pathology vs. at large.
2. The overview of foundation models is helpful, however I respectfully believe the authors are incorrect regarding foundation model capabilities. Both Virchow2 (<https://arxiv.org/abs/2408.00738>) and PLUTO (<https://arxiv.org/abs/2405.07905>) perform self-supervised pretraining at multiple resolutions, and PLUTO uses a FlexiViT which also enables the use of varied patch sizes. For architecture, Mask2Former (<https://arxiv.org/abs/2112.01527>) enables semantic segmentation adaptations on top of foundation models with ViT architecture. In all, the strongest argument for retaining an older approach is computational efficiency—and on this I certainly agree with the authors. However, I would caution them against dismissiveness toward using a foundation model approach. Embedding an entire WSI once and using multiple adaptations for downstream tasks including artifact identification is likely less expensive than embedding many times each for a different task. Thus, I still do believe that this approach is older, though I also believe (and said in the original review) that the tool itself is useful.
3. Regarding dataset size, the authors rebut comparison because of the implied comparison with self-supervised. I'm sorry to be unclear here—the comparisons in dataset size I am suggesting are with the supervised components or tasks built on top of the self-supervised models. See e.g. <https://arxiv.org/abs/2408.02859> (on the order of 1000-5000 slides) for one reference; or <https://www.nature.com/articles/s41591-024-03172-7> (on the order of 8000 slides) for another specifically doing artifact detection as one component of a computational pathology workflow. For an intended “global” use of this tool, I do think the dataset used to develop and evaluate the model remains limited which will lead to challenges in generalizability (despite significant augmentation).

In summary, I think there remain some significant limitations to the advance that this study presents, however, I do think it is a meaningful contribution to computational pathology which may be more narrow than the focus of this journal. It is challenging to evaluate this aspect without seeing the code as well. I leave this decision to the editor, and I thank the authors again for this effort and open sourcing the result!

(Remarks on code availability)

It would be helpful to see the code if the major contribution of the paper is the tool itself.

Reviewer #2

(Remarks to the Author)

I copy and paste the last comments of Dr. Asier Antoranz, my post doc, here:
The authors have satisfactorily addressed all the comments/concerns I had. I would like to congratulate the authors for their efforts.

(Remarks on code availability)

The code is provided via a github page. It's available and can be run.

Reviewer #3

(Remarks to the Author)

(Remarks on code availability)

Dear Editors and Reviewers,

We are thankful for the consideration of our work and for a plethora of constructive comments that allowed us to improve our work.

In the following, we address all comments of the Reviewers in full and provide a step-by-step explanation of additional analysis performed and changes made.

The temporary link to GrandQC repositiorium for peer-review (public; the code will be systematized upon publication, and more detailed instructions and examples will be provided for the end users):

<https://github.com/Superzlw/GrandQC>

The link to checkpoints (Google Drive; free access):

[https://drive.google.com/drive/folders/1aR7CxQLpnM_-AKD911afROG-QtpnxEVq?usp=drive link](https://drive.google.com/drive/folders/1aR7CxQLpnM_-AKD911afROG-QtpnxEVq?usp=drive_link)

The zip-archive with full code for training and experiments was submitted as a part of initial submission (188 MB) – a link was provided to us by the Editors. However, also available under the link for checkpoints.

Sincerely,

Yuri Tolkach,

on behalf of all authors

Reviewer #1 (Remarks to the Author):

In this paper, the authors propose GrandQC, a computer vision algorithm for artifact detection and segmentation algo on pathology whole-slide images (WSIs). The algorithm performs several useful functions, most notably segmenting the tissue in a slide and performing a multi-class artifact detection and segmentation. The algorithm can be considered a base for other digital pathology workflows, and could be used to do things that the authors propose such as assess hospital-specific failure modes or trends over time in digitization.

There are many useful characteristics of this study, most notably:

- release of the model as open-source
- release of prediction masks on TCGA
- the diversity of data sources used to train the model
- strong performance of the model
- careful consideration into how to do the classification of different artifacts

The authors have thought carefully about how to design this tool, and it is well-reflected in the work.

However, although this is a meaningful contribution to the field of applied computer vision in pathology, I do not think this study is of broad interest or novelty to support its acceptance at the current venue. There are several major concerns supporting this conclusion:

1. The work is not novel in its approach, it is largely similar to existing tech in the field. The main novelty is in the open-access nature of the model. This is a very nice novelty, but makes it more suitable for a more specialized venue.

Response:

- The primary goal of our work is not to develop a new technical approach but to conduct applied research with wide outreaching implications to the field. It is important to highlight that artifacts in digitized whole-slide images remain a significant issue without a proper solution. Existing tools are either ineffective or have limitations in various aspects, such as algorithm type (e.g., classification algorithms that work with larger, coarser regions), detecting only specific types of artifacts, or using non-deep learning methods with low accuracy. Our work now clearly demonstrates these shortcomings through a comparison with other available tools (HistoROI, HistoQC, PathProfiler; additional experiments recommended by Reviewer 2) where we are able to show the superiority of GrandQC over all other tools by a large margin (details in the Suppl. Tables 4-6, Suppl. Figures 11-25). Below are several examples of whole-slide image analysis using GrandQC and other tools.

Comparison with other QC Tools (HistoQC, HistoROI, PathProfiler)

Suppl. Figure 9

Comparison with other QC Tools (HistoQC, HistoROI, PathProfiler)

Suppl. Figure 13

- Additionally, we show that nearly all datasets used for training self-supervised or weakly-supervised algorithms—including foundation models—contain 5-20% artificially altered areas, which introduce biases and should be excluded. Therefore, the broad adoption of GrandQC in such studies is expected to significantly enhance the quality of the resulting algorithms.

- Our tool is, therefore, intended for widespread use by a broad audience, including researchers, pathologists, pathology institutes, and across various contexts.

2. The approach is not state-of-the-art, it is more dated than e.g. recent results showing the utility of foundation models for pathology, which use multiple orders of magnitude greater numbers of slides and patches. Again, this doesn't prove troubling for the actual application which is more straightforward and may not necessitate SOTA ML methods.

Response:

While we agree with the reviewer that foundation models are not strictly necessary in the context of our tool, we recognize that foundational encoders play a significant role in modern algorithm architectures. Below, and in the Discussion (Page 14), we offer further comments and insights into why this is the case.

Foundation models are notable for their robust feature extraction capabilities. Trained on large datasets of whole-slide images from various pathology domains, they excel at extracting feature vectors, such as those of shape [1, 1024], from images. These features can then be utilized in downstream applications, such as aggregating data from individual whole-slide images to make predictions, training linear classifiers, or feeding them into a segmentation decoder—relevant in our scenario where we are training a multi-class, pixel-wise segmentation model.

However, we believe that foundation models or encoders (e.g., UNI, Prov-GigaPath, CONCH, CTransPath, etc.) are not well-suited for multi-class semantic segmentation, at least for to this timepoint. There are three primary reasons for this:

1) **Patch Size Limitation:** Foundation models typically operate with a patch size of 224 pixels. This limited context is inadequate for tissue semantic segmentation, as it fails to capture object boundaries effectively. A well-documented empirical issue is that pixel-level predictions at the edges of patches (around 64 pixels from each side) are much less reliable for segmentation models. A common approach to improve segmentation accuracy, often used in commercial algorithms, is to focus on the central regions of patches and disregard the boundaries. With a 224-pixel patch, the reliable prediction area becomes very small.

2) **Magnification Differences:** Most foundation models are trained using an objective magnification of 20-40x (micron-per-pixel, MPP, of 0.25-0.5). In contrast, our problem requires training at 5x to 10x magnification to ensure accurate results at 5x. This provides significant computational efficiency compared to models at 20x (16 times less computation) and 40x (64 times less computation).

3) **Incompatibility with Patch Size and Network Design:** Current network implementations (e.g., U-Net or U-Net++) are designed to work with patch sizes that are powers of two (e.g., 256, 512, or 1024 pixels), allowing for multiple downsampling steps. Using other patch sizes would require redesigning the network or accepting information loss due to alignment issues. U-Net and U-Net++ are state-of-the-art for semantic segmentation and consistently deliver reliable results.

4) **Computational Load:** Reducing the patch size does not significantly reduce the computational demands for processing each patch. In fact, smaller patch sizes increase computational load and processing time—using 224-pixel patches compared to 512-pixel patches results in a 5x increase in processing time. For diagnostic, and especially algorithms (intended to quickly analyze large amounts of whole-slide images), this is unacceptable, as it would lead to tissue segmentation times of 25-30 minutes per slide.

These considerations are also why multi-class tissue semantic segmentation has not been evaluated in the original publications of state-of-the-art foundation models such as UNI and Prov-GigaPath. While these models have been applied to cell segmentation, where objects are smaller and smaller patch sizes offer a reasonable trade-off between performance and accuracy, they are not suitable for our multi-class tissue segmentation task.

3. The dataset is not particularly large or diverse in comparison to other published methods for ML in pathology, which may limit generalizability of the model (see e.g. recent results from Faisal Mahmood's lab).

This remains a useful contribution, however, due to the open source nature of the model and the other novelties in the approach.

Response:

First, we believe that comparing our fully supervised approach to the studies conducted by the Mahmood lab may not be particularly meaningful. The reviewer likely refers to the development of the UNI foundational model (or CONCH). These studies involve pre-training foundation encoders on large datasets in a self-supervised manner, which represents a fundamentally different paradigm from our approach. The models developed by the Mahmood group do not directly detect artifacts; they still require additional training on specific tasks using new datasets. Furthermore, as outlined in our previous response, there are significant limitations to these models.

Second, from the perspective of supervised learning, our training dataset is both large and diverse. It includes whole-slide images from the TCGA dataset, which is compiled from materials across 36 different institutions and represents a variety of organs. These images have been carefully curated and manually annotated by experienced pathologists to capture heterogeneous regions and contexts within individual whole-slide images.

The number of annotated slides in our dataset is also highly competitive, and we employ state-of-the-art data augmentation techniques. In numerous studies on developing clinical-grade tools for digital pathology, we have demonstrated that similar setups — or even smaller annotated datasets — are sufficient to achieve the highest levels of generalizability (Kludt et al. Cell Rep Med 2024, Giammanco et al. Mod Path 2024, Grimm et al. Mod Path 2023, Tolkach et al. NPJ Prec Oncol 2023) in independent multi-institutional validation.

I have additional questions regarding the approach taken, where some additional detail is necessary to evaluate the results. These are more minor:

- How does performance compare to pathologist performance at the tasks? I.e., do two pathologists agree on artifact areas about as much as the model agrees with each pathologist?

Response:

In our study, we evaluate the performance of the tool in comparison to ground truth produced by two human analysts: a doctoral student with a medical background who annotated the large test dataset, and an experienced board-certified pathologist who reviewed and corrected those annotations. However, we believe that interobserver agreement is less relevant or not applicable at all for artifact detection, as it typically does not involve assessment subjectivity or a second opinion. Moreover, artifact detection is not necessarily a pathologist's task, since quality control of slides is often handled by technical assistants. In this context, artifact detection differs from classical diagnostic tasks like Gleason grading or tumor subtyping, where multiple pathologists must be typically involved in validation due to the inherently subjective

nature of such evaluations (Tolkach et al., Nat Mach Intel 2020; Tolkach et al NPJ Prec Oncol 2023; Harder et al. Mod Path 2024; Kludt et al. Cell Rep Med 2024).

- Why does one need models at both 5x and 10x? If scanned at 10x, couldn't one simply pass the second layer of the WSI pyramid through the 5x model?

Response:

The reviewer's comment refers to the pyramidal structure of whole-slide images. However, this does not directly relate to having two different algorithm versions operating at different magnifications. While higher magnifications may improve segmentation or artifact detection, this benefit has a limit. A widely used magnification level for multi-class semantic segmentation in histology is 10x, offering a good balance between analysis speed and accuracy. In contrast, 20x and especially 40x magnifications are computationally expensive (as previously explained), have restricted context information in a patch, and tend to focus excessively on details, which often results in noise and false classifications.

In our specific application, we focus on the pre-analytical step applied to all whole-slide images, such as those scanned in the institute. To address this, we train three algorithm versions, each with varying accuracy. For example, the 5x magnification algorithm has slightly lower segmentation accuracy but offers significantly higher computational efficiency/lower analysis times compared to higher magnifications. This is critical for pre-analytical processing. Moreover, we demonstrate that the 5x model's performance remains highly competitive. We believe offering these model variations provides substantial value to end users, allowing them to prioritize either analysis speed or precision based on their specific needs.

As for the layers in the WSI pyramid, they can be used for computational optimization, but they are unrelated to the algorithm's working magnification. For instance, if a WSI has pyramid levels of 1x, 5x, 10x, 20x, and 40x, and the algorithm operates at 10x magnification with a patch size of 512x512 pixels, there are three potential approaches:

At 40x magnification: extract a 2048x2048 px patch > resize it to 512x512 px (now at 10x magnification) > model inference

At 20x magnification: extract a 1024x1024 px patch > resize it to 512x512 px (now at 10x magnification) > model inference

At 10x magnification: extract a 512x512 px patch > model inference

Option 3 is the most computationally efficient since no resizing is needed, and the smaller patch size reduces memory usage (reading from the WSI and manipulating it in RAM is more efficient). This method can save 5-10% of CPU resources, enabling faster image processing compared to option 1.

- More broadly, is "10x" the appropriate measurement of a length scale here? A more precise length measurement would be the edge length of a pixel from the scan, e.g. 1.0 micron per pixel (mpp).

Response: Exactly, we use the term "magnification" primarily for ease of understanding. However, in our Methods section and throughout all scripts, we consistently refer to the MPP (microns per pixel) parameter instead of magnification. For reference, an MPP of roughly 0.25 corresponds to 40x magnification, 0.5 to 20x, and 1.0 to 10x. This approach provides more precision in describing the resolution without relying on traditional magnification terms.

- What does it mean that dense annotations were created "where possible?" Was there a specific strategy for choosing where to densely annotate?

Response: We now clarify this in the Methods section. In most cases, we utilized dense annotations. The exception to this rule involves regions with numerous small artifacts (typically dark spots), where the surrounding areas are a mix of small out-of-focus regions and unaffected regions, where it is difficult to establish a ground truth. In these cases, we applied sparse annotations to mark the artifacts while leaving the surrounding areas unannotated.

- Clinical validation typically refers to evaluation of a tool for a specific purpose with predefined acceptance criteria. The CV shown here is a useful evaluation of the model performance, but does not involve the rigor of a clinical validation (e.g. power calculations, AC, etc.). "Model evaluation" may be a more appropriate term, but the authors should be careful in describing the evaluation work performed.

Response: We use now the term "Evaluation of the algorithm in a clinical context" to avoid any ambiguity and ensure clarity in the interpretation.

###

Reviewer #2 (Remarks to the Author):

Summary

The authors present grandQC, a quality control model for tissue and artifact detection in HE stainings. Their model is developed using a large multi-organ and multi-institutional dataset. They extensively test and validate its performance and show the validity of the model for different applications including benchmarking of staining quality for pathology departments, and assessing staining quality over time. The manuscript is suitable for the journal and will be of significance to the field. The open-sourceness of the model and the annotations will be of significant help for further developments. Having said that, there are a number of points that require further work before the manuscript is ready for publication.

Response: We appreciate the reviewer's positive reception of our work and their valuable, constructive feedback, which has helped us elevate the quality of our research.

Major comments

1- The relevance of artifact removal or artifact misclassification in downstream analyses is mentioned several times in the text. Examples include abstract (lines 49-50), results (lines 257-259), among others. The authors provide several references but there is no demonstration of how grandQC improves downstream analysis algorithms. The manuscript would gain significant impact if this potential improvement is quantified in a couple of examples.

Response:

We conducted additional experiments to explore and demonstrate how GrandQC positively influences downstream analyses. To do this, we employed two clinical-grade, multi-class segmentation algorithms (for colorectal cancer, Griem J. et al., Mod Path 2023, and lung cancer, Kludt C. et al., Cell Rep Med 2024) along with one single-cell detection/classification algorithm. These algorithms were previously developed and extensively validated by us. We evaluated three distinct use cases/situations to highlight the benefits of GrandQC. The details of technical implementation of these

analyses can be also found in **Methods: “Evaluation of downstream algorithm performance in context of quality control”**. Below is the summary of the results:

1. Artifacts as a Source of False Positive Tumor Classifications in Benign Tissue.

One cause of false-positive tumor classifications in benign tissue is the presence of artifacts. Using test datasets independent of the algorithm’s training data (as described in each publication), we extracted regions with benign tissue containing artifacts. For colorectal cancer (WSI/case n=30) and lung cancer domains (WSI/case n=40), we extracted 33 and 105 benign tissue regions with artifacts, respectively. Our results show that 7/33 (21.2%) colorectal regions and 19/105 (18.1%) lung regions were falsely classified as tumors due to these artifacts. However, GrandQC successfully detected these artifacts, preventing these misclassifications at the pre-analytical stage in all regions. The summary and examples are provided in **Figure 8A** (new) and below. Further examples and more detailed statistics per artifact type are presented in **Suppl. Figure 8**.

A GrandQC: downstream performance, use case 1

Preventing false positive tumor misclassifications in regions with benign tissue

GrandQC effectively preventing false positive tumor detections (two clinical-grade multi-class AI tools for lung and colorectal specimens). GrandQC detects the artificially changed regions and masks them from downstream processing. **[Figure 8A]**

2. Artifacts Significantly Impact Segmentation Quality, Leading to False Negative Tumor Classifications (High Clinical Relevance).

In this case, we reanalyzed the test datasets from abovementioned publications but focused on tumor regions that were precisely manually annotated by experts during the original studies. These regions include epithelial tumor tissue and tumor-associated structures (e.g., tumor stroma, necrotic debris, and tertiary lymphoid structures for lung cancer; tumor stroma, necrotic debris, and mucin for colorectal cancer). Since these annotated regions were selected to avoid significant artifacts, we artificially introduced syntethic out-of-focus (OOF) artifacts into portions of the regions to evaluate their impact (principle is similar to training dataset). The final datasets

included 126 tumor regions (2,016 tiles) for lung cancer and 121 tumor regions (1,936 tiles) for colorectal cancer from the WSI/patient n=40 (lung cancer), n=30 (colorectal cancer).

Our analysis revealed that OOF artifacts reduced segmentation quality (measured by the Dice score) and significantly decreased sensitivity of tumor detection, which is critical in a clinical context. GrandQC significantly improved these parameters by effectively detecting artificially altered regions. However, it is important to note and understand (for end-users) that while GrandQC detects and masks artifacts preventing misclassifications, it does not enhance the algorithm's ability to detect structures in the affected areas. Therefore, thresholds may be needed to prompt pathologists to conduct additional reviews when large artifact areas are detected in certain context, as these may obscure important findings. We believe the selection of these thresholds is subjective and should be aligned with the pathologist's experience and working style. Detailed results of this analysis are provided below (and in a new Figure 8B, Suppl. Figures 9,10).

B GrandQC: downstream performance, use case 2

Improving segmentation accuracy in tumor region

Lung AI tool					Colorectal AI tool				
		Dice	SENS	SPEC		Dice	SENS	SPEC	
Tumor	Baseline	0.919	0.916	0.931	Baseline	0.796	0.821	0.818	
	with OoF	0.707	0.662	0.949	with OoF	0.434	0.404	0.914	
	after GrandQC	0.884	0.873	0.964	after GrandQC	0.714	0.710	0.930	
Tumor stroma	Baseline	0.843	0.841	0.951	Baseline	0.443	0.356	0.978	
	with OoF	0.659	0.635	0.945	with OoF	0.259	0.201	0.980	
	after GrandQC	0.798	0.804	0.969	after GrandQC	0.386	0.303	0.993	
Necrosis	Baseline	0.919	0.944	0.963	Baseline	0.528	0.677	0.944	
	with OoF	0.687	0.797	0.917	with OoF	0.323	0.412	0.965	
	after GrandQC	0.847	0.867	0.975	after GrandQC	0.467	0.602	0.960	
TLS	Baseline	0.848	0.926	0.992	Baseline	0.909	0.984	0.960	
	with OoF	0.787	0.880	0.971	with OoF	0.565	0.515	0.988	
	after GrandQC	0.835	0.940	0.991	after GrandQC	0.881	0.874	0.994	

GrandQC improves segmentation accuracy in tumor regions. [Figure 8B]

Impact of out-of-focus artifact on downstream tasks – Lung AI tool (Kludt et al. Cell Rep Medicine 2024)

Suppl. Figure 9

Lung AI tool: processing OOF artifacts with and without QC by GrandQC

Impact of out-of-focus artifact on downstream tasks – Colorectal AI tool (Griem et al. Mod Path 2023)

Suppl. Figure 10

Colorectal AI tool: processing OOF artifacts with and without QC by GrandQC

3. Artifacts cause misclassifications in single cell detection and classification, which can be prevented

Single-cell detection and classification algorithms are particularly vulnerable to artifacts due to the small size and subtle features of the cells being analyzed. We provide examples of this using a single-cell detection/classification model trained with the StarDist algorithm and the Lizard dataset. Our findings show that, by incorporating GrandQC in the pre-analytical stage and masking artifact-affected regions, false cell detections and misclassifications might be effectively prevented.

The examples of this are provided below (and in new Figure 8C).

C GrandQC: downstream performance, use case 3
Preventing false single-cell detections/classifications

GrandQC improves single-cell detection and classification in tumor region. In this particular example tissue fold and accompanying out-of-focus result in false cell detections that are being classified as tumor cells (critical misclassification) [Figure 8C]

We summarize these findings in Results and provide information about methods used in this analysis in Methods.

2- The authors do not compare the performance of GrandQC with any other method. I agree that other methods might not be directly comparable, but comparisons can be made. They mention a number of these in their manuscript: HistoQC, PathProfiler, HistoROI, etc.). Reporting the results obtained in the corresponding manuscripts might not be relevant as the performance metrics might be dataset dependant. It is necessary that whenever applicable, the authors compare grandQC with other SOTA methods (for the type of artifacts that are applicable).

Response:

We compare GrandQC directly with three other tools—HistoQC, PathProfiler, and HistoROI—using a large, four-organ test dataset, and demonstrate that GrandQC achieves significantly higher performance. Since each tool handles artifacts differently, we apply specific aggregation strategies for individual artifact classes to enable a direct comparison. Additionally, we analyze the performance by separating the dataset into different organs, providing more detailed insights based on domain-specific performance (e.g., some tools, like PathProfiler, were developed for a specific organ type, prostate). The metrics are provided in Supplementary Tables 4–6, and extensive visual examples can be found in Supplementary Figures 11–25 (including original images in higher resolution) and provided below. The details on implementation of comparisons can be found in Methods.

Suppl. Table 4. Comparison of HistoROI and GrandQC tools

Dice Score		HistoROI	GrandQC
Colon	Tissue w/o artifacts	0.757	0.874
	Artifacts	0.459	0.682
Breast	Tissue w/o artifacts	0.661	0.840
	Artifacts	0.584	0.747
Kidney	Tissue w/o artifacts	0.499	0.932
	Artifacts	0.215	0.546
Prostate	Tissue w/o artifacts	0.697	0.917
	Artifacts	0.390	0.584
Pen markings	Pen markings	0.818	0.984

Suppl. Table 5. Comparison of PathProfiler and GrandQC tools

Dice Score		PathProfiler	GrandQC
Colon	Tissue w/o artifacts	0.741	0.874
	FOLD	0.057	0.801
	OOF	0.036	0.903
	Other Artifacts	0.364	0.515
Breast	Tissue w/o artifacts	0.656	0.840
	FOLD	0.132	0.776
	OOF	0.043	0.908
	Other Artifacts	0.176	0.533
Kidney	Tissue w/o artifacts	0.644	0.932
	FOLD	0.113	0.687
	OOF	0.000	0.807
	Other Artifacts	0.091	0.482
Prostate	Tissue w/o artifacts	0.758	0.917
	FOLD	0.111	0.727
	OOF	0.017	0.671
	Other Artifacts	0.190	0.599

Pen markings	Pen markings	0.665	0.984
--------------	-------	-------

Suppl. Table 6. Comparison of HistoQC and GrandQC tools

	Dice Score	HistoQC	GrandQC
Colon	Tissue w/o artifacts	0.448	0.871
	Artifacts + background	0.122	0.716
Breast	Tissue w/o artifacts	0.491	0.829
	Artifacts + background	0.050	0.808
Kidney	Tissue w/o artifacts	0.431	0.929
	Artifacts + background	0.258	0.691
Prostate	Tissue w/o artifacts	0.439	0.913
	Artifacts + background	0.052	0.817
Pen markings	Pen markings	0.893	0.984

Comparison with other QC Tools (HistoQC, HistoROI, PathProfiler)

Suppl. Figure 11

Comparison with other QC Tools (HistoQC, HistoROI, PathProfiler)

Artifacts

Suppl. Figure 12

Comparison with other QC Tools (HistoQC, HistoROI, PathProfiler)

Artifacts

Suppl. Figure 13

Comparison with other QC Tools (HistoQC, HistoROI, PathProfiler)

Artifacts

Suppl. Figure 14

Comparison with other QC Tools (HistoQC, HistoROI, PathProfiler)

Artifacts

Suppl. Figure 15

3- In line with the previous point, the authors report the results of their model per artifact type but not per pathology center. Are there significant differences in predictive performance of GrandQC based on pathology department?

Response:

We evaluated GrandQC's performance in a clinical context using materials from various pathology centers (Figure 6). All slides from these centers were thoroughly analyzed (does not apply to Figure 7/TCGA, where, due to the large number of slides, we only screened for obvious issues). Experienced analysts (YT, ZW, AS) reviewed the results of multi-center study (Figure 6), confirming that GrandQC's performance was adequate and comparable to that seen in our large four-organ test dataset. This

can also be inferred indirectly from the distribution of artificially altered areas across different centers, which followed a consistent pattern.

Outlier performance was identified in only one center, where an older version of the 3D HISTECH scanner resulted in a specific scanning quality issue. In this case, GrandQC overcalled certain true-positive dark spot artifacts (e.g., dust particles and fingerprints), incorrectly including significant portions of unaffected neighboring tissue in the artifact segmentation region.

A more detailed comparison of performance across centers, including statistical metrics such as the Dice score, would require extensive manual annotation. Given that manual annotations in this study took over 12 months, this falls outside the scope of our current research. However, we believe that the multi-level validation provided here is sufficient to draw robust conclusions about GrandQC's performance.

4- The added value of the tissue detection module in the artifact detection module is unclear. The authors should perform an ablation study (or similar) to check the gain in performance (if any).

Response:

The tissue detection module is not designed to enhance the accuracy of the artifact detection module. Instead, tissue and artifact detection are envisioned as two distinct tasks.

First, in many cases, potential end users (such as researchers) may only need tissue detection for their downstream applications. GrandQC provides high-quality tissue detection within seconds by using a low-resolution analysis (microns per pixel/MPP of 10, which roughly corresponds to a 1x magnification). Detailed analysis times, shown in Figure 4C, are highly competitive, with a mean of 0.45 seconds for resection slides and 0.32 seconds for biopsy slides with multiple tissue levels.

Second, while the tissue detection module doesn't improve segmentation accuracy, it does offer a significant computational performance boost to the artifact detection algorithm. The artifact detection algorithm, which operates at a much higher resolution, includes "background" as a class and can detect background at the high resolution level (e.g., small background areas in the lumen of small glands), which isn't possible at the 1x magnification used by the tissue detection algorithm. According to our performance analysis dataset (Figure 4C), the average background area in a whole-slide image of a resection specimen is about 64% (approximately 450,000,000 μm^2), and 84% (600,000,000 μm^2) in biopsy specimens with multiple tissue levels. The tissue detection module can identify these background regions in under 0.5 seconds for both resection and biopsy cases, while the artifact detection module would take an additional 59 seconds (on average, for the 5x version) for resection and 34 seconds (on average) for biopsy cases.

These facts highlight the crucial role of the tissue detection module.

5- Methods, lines 155-156: the authors mention that the test dataset was used to validate the tissue detection algorithm. Is this correct? Why was this dataset not used to validate the artifact detection algorithm? The dataset was specifically developed for the generation of artifactual areas.

Response:

Due to the mistake in initial reading of your comment, we thought that your question refers to the fact why the generated large multi-organ test dataset was not used to

test the tissue detection module (it was, of course, extensively used for testing the artifact segmentation algorithm). Your actual comment refers to a simple typo, which was corrected.

As our initial reading of your comment was different, we provide here additional validation of the GrandQC's tissue detection algorithm on that dataset:

For now, we have annotated a larger portion of the test dataset (WSI n=100), which includes manually corrected annotations using QuPath automation. We ensured roughly equal representation of four organ/specimen types (colorectal, breast, prostate, and kidney). Additionally, we "overfitted" the dataset to slides with all types of artifacts. While the distribution of artifacts in these slides exceeds what is typically seen in real-world scenarios, this approach is intended to exacerbate any potential issues the tissue detection module might encounter, particularly those related to out-of-focus areas, air bubbles, and pen marks.

Despite this challenging setup, our validation experiments demonstrate a very high segmentation accuracy, regardless of organ type with an average Dice score of 0.957. The results are presented below and in Suppl. Table 3.

Suppl. Table 3. Evaluation of segmentation accuracy for tissue detection algorithm

Number of slides from different organs with different artifacts					
Validation Dataset	Colon	Breast	Prostate	Kidney	Total
OOF / glue drops	5	5	5	5	20
Air bubbles	2	2	3	3	10
Pen markings	2	2	3	3	10
Folds	15	15	15	15	60
Total	24	24	26	26	100

Dice Score	Colon	Breast	Prostate	Kidney
Tissue	0.938	0.902	0.968	0.922
Background	0.987	0.969	0.979	0.988

6- The authors mention that different encoder and decoder architectures were tested. They also mention that full details on the training procedures in the supplementary methods. However, nothing is mentioned regarding the architectures tested nor results are reported for different hyperparameter choices. Nothing is mentioned regarding the number of epochs, loss functions, or anything of the like. A much more thorough description of the training procedures as well as the obtained results (also for the non-optimal configurations) is required.

Response:

We now provide the summary of all the development steps related to selection of optimal architecture and algorithm hyperparameter fine-tuning in the Suppl. Table 1 (also provided below in the low resolution).

Suppl. Table 1. Algorithm development steps (finding optimal architecture and hyperparameters)

Default Parameters									
	Epochs	Patch Size	Optimizer	Initial Learning Rate					
	64	512	Adam	0.0005					
Comparison: Different ways to generate Out of Focus									
Encoder	Decoder	MPP	Batch Size	Loss	Out of Focus Generation	Dice score of OOF	Overall Dice Score		
EfficientNet-B0	UNet++	1.5	21	CrossEntropy with classes weight	Gaussian Blur with equal percentage for each kernel size	0.849	0.754		
EfficientNet-B0	UNet++	1.5	21	CrossEntropy with classes weight	90% Gaussian Blur with kernel sizes of 3 and 5	0.835	0.747		
EfficientNet-B0	UNet++	1.5	21	CrossEntropy with classes weight	90% Gaussian Blur with kernel sizes of 3 and 5	0.854	0.766		
Comparison: Different Encoder									
Encoder	Decoder	MPP	Batch Size	Loss	Out of Focus Generation	Overall Dice Score			
ResNet50	UNet	2	24	CrossEntropy with classes weight	90% Gaussian Blur with kernel sizes of 3 and 5	0.712			
EfficientNet-B0	UNet	2	21	CrossEntropy with classes weight	90% Gaussian Blur with kernel sizes of 3 and 5	0.804			
EfficientNet-B1	UNet	2	21	CrossEntropy with classes weight	90% Gaussian Blur with kernel sizes of 3 and 5	0.682			
VIT	Segmenter	2	8	CrossEntropy without classes weight	90% Gaussian Blur with kernel sizes of 3 and 5	0.558			
Comparison: Different Loss Function and Decoder									
Encoder	Decoder	MPP	Batch Size	Loss	Out of Focus Generation	Overall Dice Score			
ResNet50	UNet++	2	24	Lovasz-Softmax Loss + CrossEntropy	90% Gaussian Blur with kernel sizes of 3 and 5	0.616			
ResNet50	UNet++	2	24	BoundaryLoss + CrossEntropy	90% Gaussian Blur with kernel sizes of 3 and 5	0.584			
ResNet50	UNet++	2	24	0.6Focal + 0.4Dice	90% Gaussian Blur with kernel sizes of 3 and 5	0.689			
ResNet50	UNet++	2	24	CrossEntropy with classes weight	90% Gaussian Blur with kernel sizes of 3 and 5	0.728			
ResNet50	UNet++	2	24	Lovasz-Softmax Loss	90% Gaussian Blur with kernel sizes of 3 and 5	0.627			
ResNet50	UNet	2	24	Focal	90% Gaussian Blur with kernel sizes of 3 and 5	0.704			
ResNet50	UNet	2	24	Dice Loss	90% Gaussian Blur with kernel sizes of 3 and 5	0.710			
ResNet50	UNet	2	24	Lovasz Loss	90% Gaussian Blur with kernel sizes of 3 and 5	0.691			
Comparison: Different Batch Size and Decoder									
Encoder	Decoder	MPP	Batch Size	Loss	Out of Focus Generation	Oversampling	Overall Dice Score		
EfficientNet-B0	UNet++	1.5	14	CrossEntropy with classes weight	90% Gaussian Blur with kernel sizes of 3 and 5	15k	0.778		
EfficientNet-B0	UNet++	1.5	21	CrossEntropy with classes weight	90% Gaussian Blur with kernel sizes of 3 and 5	15k	0.778		
EfficientNet-B0	UNet++	1.5	28	CrossEntropy with classes weight	90% Gaussian Blur with kernel sizes of 3 and 5	15k	0.788		
EfficientNet-B0	DeepLabv3+	1.5	14	CrossEntropy with classes weight	90% Gaussian Blur with kernel sizes of 3 and 5	15k	0.810		
EfficientNet-B0	DeepLabv3+	1.5	21	CrossEntropy with classes weight	90% Gaussian Blur with kernel sizes of 3 and 5	15k	0.797		
EfficientNet-B0	UNet	1.5	14	CrossEntropy with classes weight	90% Gaussian Blur with kernel sizes of 3 and 5	15k	0.783		
EfficientNet-B0	UNet	1.5	21	CrossEntropy with classes weight	90% Gaussian Blur with kernel sizes of 3 and 5	15k	0.773		
Comparison: Different Amounts of Oversampling and Decoder									
Encoder	Decoder	MPP	Batch Size	Loss	Out of Focus Generation	Oversampling	Overall Dice Score		
EfficientNet-B0	UNet++	1.5	14	CrossEntropy with classes weight	90% Gaussian Blur with kernel sizes of 3 and 5	10k	0.811		
EfficientNet-B0	UNet++	1.5	14	CrossEntropy with classes weight	90% Gaussian Blur with kernel sizes of 3 and 5	13k	0.793		
EfficientNet-B0	UNet++	1.5	14	CrossEntropy with classes weight	90% Gaussian Blur with kernel sizes of 3 and 5	15k	0.778		
EfficientNet-B0	UNet++	1.5	21	CrossEntropy with classes weight	90% Gaussian Blur with kernel sizes of 3 and 5	10k	0.807		
EfficientNet-B0	UNet++	1.5	21	CrossEntropy with classes weight	90% Gaussian Blur with kernel sizes of 3 and 5	15k	0.778		
EfficientNet-B0	UNet++	1.5	21	CrossEntropy with classes weight	90% Gaussian Blur with kernel sizes of 3 and 5	22k	0.780		
EfficientNet-B0	UNet	1.5	21	CrossEntropy with classes weight	90% Gaussian Blur with kernel sizes of 3 and 5	15k	0.794		
EfficientNet-B0	UNet	1.5	21	CrossEntropy with classes weight	90% Gaussian Blur with kernel sizes of 3 and 5	22k	0.794		
Comparison: Different Weight of Out of Focus									
Encoder	Decoder	MPP	Batch Size	Loss	Out of Focus Generation	Oversampling	Weights Principle	Dice score of OOF	Overall Dice Score
EfficientNet-B0	UNet	1.5	21	CrossEntropy with classes weight	90% Gaussian Blur with kernel sizes of 3 and 5	15k	OOF * 1.1	0.856	0.823
EfficientNet-B0	UNet	1.5	21	CrossEntropy with classes weight	90% Gaussian Blur with kernel sizes of 3 and 5	15k	OOF * 1.2	0.858	0.801
EfficientNet-B0	UNet	1.5	21	CrossEntropy with classes weight	90% Gaussian Blur with kernel sizes of 3 and 5	15k	OOF * 1.3	0.857	0.765
EfficientNet-B0	UNet	1.5	21	CrossEntropy with classes weight	90% Gaussian Blur with kernel sizes of 3 and 5	15k	OOF * 1.5	0.834	0.793
Comparison: Different methods for dealing with class imbalance									
Encoder	Decoder	MPP	Batch Size	Loss	Out of Focus Generation	Oversampling	Weights Principle	Overall Dice Score	
EfficientNet-B0	UNet	1.5	21	CrossEntropy with classes weight	90% Gaussian Blur with kernel sizes of 3 and 5	15k	-	0.766	
EfficientNet-B0	UNet	1.5	21	CrossEntropy with classes weight	90% Gaussian Blur with kernel sizes of 3 and 5	15k	-	0.794	
EfficientNet-B0	UNet	1.5	21	CrossEntropy with classes weight	90% Gaussian Blur with kernel sizes of 3 and 5	15k	Global + Oversampling	0.760	
EfficientNet-B0	UNet	1.5	21	CrossEntropy with classes weight	90% Gaussian Blur with kernel sizes of 3 and 5	27k	Global + Oversampling	0.786	
Final Models with different magnifications									
Encoder	Decoder	MPP	Batch Size	Loss	Out of Focus Generation	Oversampling	Weights Principle	Overall Dice Score	
EfficientNet-B0	UNet	2	21	CrossEntropy with classes weight	90% Gaussian Blur with kernel sizes of 3 and 5	10k	-	0.785	
EfficientNet-B0	UNet	1.5	21	CrossEntropy with classes weight	90% Gaussian Blur with kernel sizes of 3 and 5	27k	Global + Oversampling	0.808	
EfficientNet-B0	UNet	1	21	CrossEntropy with classes weight	90% Gaussian Blur with kernel sizes of 3 and 5	15k	-	0.824	

Importantly, the development process involved extensive rounds of visual performance reviews conducted by experts, rather than relying solely on metrics. Consequently, Dice Scores and other statistical metrics were not the sole factors influencing decisions. This is particularly relevant in cases (see Suppl. Table 1) where an algorithm version with a higher average Dice score was rejected based on visual evaluation.

7- The authors mention that during training, the air bubbles and the slide edge artifacts are merged due to their visual similarity. The same is true for dark spot and foreign object artifacts. They also mention that most misclassification problems happen between artifact types. However, they claim several times in the text that the multiclass nature of their model allows for flexibility. This is particularly noticeable when reporting the issue with pigmented tissue in melanoma slides, saying that the air bubble class could be disregarded (results, lines 264-265). Those two aspects seem a bit contradictory: sometimes the multiclass segmentation nature of the model is emphasized highlighting the relevance of getting class-level predictions; other times, they say that having the misclassifications between artifact types is not relevant. The manuscript would benefit from coherence in this matter.

Response: Thank you for mentioning this. This refers to the section “Validation of artifact detection algorithm” and the claim that inter-artifact misclassifications are not relevant was removed.

8- Results, lines 218-223: The authors present a custom score for tissue detection quality based on subjective visual evaluation by two experienced analysts. This custom score's value is limited without comparison to other methodologies in a blind setting.

Response:

The primary goal of this subanalysis was to determine, in a clinical setting, how many slides would pass the quality check after tissue detection by GrandQC (which was 100% for both analysts). However, to introduce a quantitative element, we decided to implement a rating scale. The intention of this analysis is not to compare different methods. Additionally, we address the limitations of this approach for comparing methodologies in the Limitation section of the Discussion. We also refer to the new analysis of tissue detection accuracy (see your comment above).

Minor comments

1- Abstract, lines 63-64: the authors mention that the segmentation of artifacts and tissue without artifacts (dice score 0.919-0.938). This is misleading as those values are only for the areas without artifacts. The reported dice scores for the artifacts themselves in Figure 4.B is significantly lower (0.325-0.528 for Dark spot & foreign). The result is reported in the same way in the results (lines 237-238).

Response:

This refers indirectly to Reviewer comment 7 about inter-artifact misclassifications which are the reason for lower Doce scores as far as *artifact* segmentation is concerned. However, the segmentation accuracy in *artifact-free tissue* provides a reliable estimate of the overall performance of the tool in detecting artifacts. We have clarified this point in both the Abstract and the Manuscript.

2- The authors mention that the annotations were performed manually. However, in the methods, they describe that the tissue segmentation annotations were automated in QuPath. Please correct.

Response: Indeed, the annotations for tissue detection were automated in QuPath but fully corrected manually in each case. We make this point clear in Results section.

3- The authors provide three models at 5X, 7X, and 10X magnification. Since the slides were scanned at 40X magnification, each pixel at 5X represents 64 pixels at 40X. This discrepancy might explain issues with detecting small artifacts. It raises the question of whether training at higher magnification could improve artifact detection in those cases. Downscaling the images could also impact the detection of out-of-focus (OOF) artifacts. It would be nice to check if this is the case.

Response:

While we understand the reasoning behind training the algorithm at higher magnifications, we believe this approach is not computationally efficient. GrandQC is designed to be a pre-analytical step that can quickly process a large number of slides. In numerous previous studies published stemming from our research group (AI tools for prostate, colorectal, oesophageal/gastric, lung specimens, lymph node metastasis detection), we demonstrated that 10x magnification is optimal for developing diagnostic digital pathology tools (normally involving much more complex diagnostic problems, such as multi-class tissue segmentation with morphological overlaps between classes etc., e.g. > 10 tissue classes, compared to artifact detection problem) with neither tool required higher magnification to date. This corresponds well to evidence from the published studies of other groups. Higher magnifications (e.g., 20x) not only result in intolerable analysis times (>10-15 minutes/slide) but also produce poorer segmentation results due to less surrounding context and more fine-grained details, which often act as noise for deep learning algorithms. Additionally, the algorithms trained at higher resolution might struggle in differentiating OOF regions from effects related to thicker sections, which may appear blurry but, in our view, should not be flagged as OOF (see examples below).

Below are several examples that, although not considered OOF, might become a distraction at higher resolutions:

However, we do perform additional trainings to address this comment of Reviewer experimentally. As we do not intend to release these models (based on above mentioned considerations, especially that of computational inefficiency), we did not

plan to perform any extensive fine-tuning or visual validation by experts – which proven to be extremely important for development process (e.g., models with same metrics performed very differently during visual validation). Although for certain classes we see similar results as by 5/7/10x versions, these metrics should be interpreted with extreme caution due to inter-artifact misclassification, different number of patches and their size, etc. We don't see improvement in OOF detection at the level of these metrics.

Version	Tissue without artifacts	Background	Fold	Dark spot & Foreign	Pen markings	Air Bubble & Edge	OoFocus	Average Dice
15x (MPP 0.67)	0.902	0.783	0.815	0.544	0.948	0.644	0.810	0.778
20x (MPP 0.5)	0.893	0.791	0.825	0.525	0.935	0.882	0.842	0.813

4- Methods, lines 185-186: the reference regarding supplementary figure 1 appears to be incorrect. The authors are referring to the details on the datasets but supplementary figure 1 refers to the artificial generation of OOF areas.

Response: Was corrected to Suppl. Table 2.

5- Results, line 230: The authors refer to supplementary figures 4 and 5 for the comparative of the different versions. However, the correct figures seem to be supplementary figures 3 and 4. Additionally, these figures do not show the ground truth and the provided magnification of the figures makes their interpretation challenging. Moreover, at least with the provided example, there seems to be quite some disagreement between all three models. A concordance study should be conducted to evaluate the differences in predicted areas between the models.

Response: We have corrected the reference to the supplementary figures.

The images shown are just examples from one of the clinical datasets and do not include ground truth data.

Importantly, we identified the mistake in the legend of both these images – the upper mask corresponds to 10x version, mask in the middle corresponds to the 7x version, and the lower mask – to 5x version.

Given this correction, the resulting artifact detections very well correspond to the Dice score trends for single artifact classes that can be seen in the Figure 4B and well illustrate the differences between the versions for end users. Additionally, we provide a more detailed comparative analysis of these images at higher resolutions (see Supplementary Figures 5-6).

There are several other potential sources of discrepancies:

- 1) Stochastic nature of patch coordinates during whole-slide image analysis.
- 2) It is well known that segmentation models tend to be less reliable in the border regions of patches (approximately 10-15%) due to the lack of surrounding context, whereas the central regions yield more accurate predictions. At different magnifications, patches are placed differently within a whole-slide image, that contributes to discrepancies across model versions.

While a common solution to this issue is to overlap patches by 20-25% during whole-slide inference and stitch only the central regions of the resulting masks (a method that significantly improves segmentation quality at the cost of increased

computational resources), we leave this fine-tuning to the end users. Our approach maintains a balance between more accurate segmentation and the need for faster analysis, which is crucial for a pre-analytical tool.

3) Moreover, for OOF artifacts for algorithm “seeing” transition between OOF region and tissue without artifact rather than having a full patch involved into artificial area empirically significantly improves the segmentation.

Most of these limitations apply to all deep learning-based tools.

6- Typographic error in line 119: AA.

Response: Was corrected.

###

Reviewer #3 (Remarks to the Author):

I co-reviewed this manuscript with one of the reviewers who provided the listed reports. This is part of the Nature Communications initiative to facilitate training in peer review and to provide appropriate recognition for Early Career Researchers who co-review manuscripts.”

Response: Thank you for your valuable contribution to review of this manuscript.

Dear Editors and Reviewers,

We sincerely thank you for the valuable and insightful comments, as well as for the positive feedback on our work.

In the following, we address the comments of the Reviewers point-by-point below.

Reviewer #1 (Remarks to the Author):

1. It is very helpful to compare this method to other approaches, however, this does also highlight that the modeling approach is not radical, rather a notable step forward from prior approaches. Again, the key contribution of a tool that is open-source, performant, and compute-efficient is meaningful, but more so within the field of computational pathology vs. at large.
2. The overview of foundation models is helpful, however I respectfully believe the authors are incorrect regarding foundation model capabilities. Both Virchow2 (<https://arxiv.org/abs/2408.00738>) and PLUTO (<https://arxiv.org/abs/2405.07905>) perform self-supervised pretraining at multiple resolutions, and PLUTO uses a FlexiViT which also enables the use of varied patch sizes. For architecture, Mask2Former (<https://arxiv.org/abs/2112.01527>) enables semantic segmentation adaptations on top of foundation models with ViT architecture. In all, the strongest argument for retaining an older approach is computational efficiency—and on this I certainly agree with the authors. However, I would caution them against dismissiveness toward using a foundation model approach. Embedding an entire WSI once and using multiple adaptations for downstream tasks including artifact identification is likely less expensive than embedding many times each for a different task. Thus, I still do believe that this approach is older, though I also believe (and said in the original review) that the tool itself is useful.

Response:

- **Thank you for the feedback. We also agree that using the selected approach is optimal for the task we have.**

3. Regarding dataset size, the authors rebut comparison because of the implied comparison with self-supervised. I'm sorry to be unclear here—the comparisons in dataset size I am suggesting are with the supervised components or tasks built on top of the self-supervised models. See e.g. <https://arxiv.org/abs/2408.02859> (on the order of 1000-5000 slides) for one reference; or <https://www.nature.com/articles/s41591-024-03172-7> (on the order of 8000 slides) for another specifically doing artifact detection as one component of a computational pathology workflow. For an intended “global” use of this tool, I do think the dataset used to develop and evaluate the model remains limited which will lead to challenges in generalizability (despite significant augmentation).

Response:

- Thank you for clarifying your point on dataset size and for providing relevant references. In the first referred publication the authors develop a tool for slide level predictions – something completely different from our task of semantic multi-class segmentation and could not be compared to our methodology. In the second publication, the number of slides used is highly questionable. In our previous publications (cited in the paper, e.g. Kludt et al. 2024, Griem et al. 2023, Giammanco et al. 2024, Tolkach NPJ Prec Oncol 2023) we show that models developed using 300-400 slides generalize very effectively to unseen data from other institutions (sometimes 5 different institutes) – we have never seen any problems with generalizability as far as training methodology accounts for this issue.

Reviewer #1 (Remarks on code availability)

1. It would be helpful to see the code if the major contribution of the paper is the tool itself.

Reviewer #2 (Remarks to the Author)

1. I copy and paste the last comments of Dr. Asier Antoranz, my post doc, here:
The authors have satisfactorily addressed all the comments/concerns I had. I would like to congratulate the authors for their efforts.

Reviewer #2 (Remarks on code availability):

1. The code is provided via a github page. It's available and can be run.

Reviewer #3 (Remarks to the Author)
